# SEL1L-HRD1 interaction is required to form a functional HRD1 ERAD complex

Liangguang Leo Lin [1], Huilun Helen Wang[1], Brent Pederson[2], Xiaoqiong Wei [1], Mauricio Torres[2], You Lu [2,5], Zexin Jason Li[1], Xiaodan Liu[3,6], Hancheng Mao [2], Hui Wang[1], Linyao Elina Zhou[1], Zhen Zhao [3], Shengyi Sun [4] ✉ & Ling Qi [1] ✉

The SEL1L-HRD1 protein complex represents the most conserved branch of endoplasmic reticulum (ER)-associated degradation (ERAD). Despite recent advances in both mouse models and humans, in vivo evidence for the importance of SEL1L in the ERAD complex formation and its (patho-)physiological relevance in mammals remains limited. Here we report that *SEL1L* variant *p.Ser658Pro* (*SEL1L^{S658P}*) is a pathogenic hypomorphic mutation, causing partial embryonic lethality, developmental delay, and early-onset cerebellar ataxia in homozygous mice carrying the bi-allelic variant. Biochemical analyses reveal that *SEL1L^{S658P}* variant not only reduces the protein stability of SEL1L, but attenuates the SEL1L-HRD1 interaction, likely via electrostatic repulsion between SEL1L F668 and HRD1 Y30 residues. Proteomic screens of SEL1L and HRD1 interactomes reveal that SEL1L-HRD1 interaction is a prerequisite for the formation of a functional HRD1 ERAD complex, as SEL1L is required for the recruitment of E2 enzyme UBE2J1 as well as DERLIN to HRD1. These data not only establish the disease relevance of SEL1L-HRD1 ERAD, but also provide additional insight into the formation of a functional HRD1 ERAD complex.

In eukaryotes, approximately 30% of all newly synthesized proteins pass through the endoplasmic reticulum (ER), where they undergo folding and maturation[1]. Although a fraction of nascent proteins fails to fold properly, evolutionary biology has accounted for this, as many of these proteins are ultimately disposed of by a quality-control process known as ER-associated degradation (ERAD)[2–4]. While ERAD has been implicated in over 70 human diseases[5], our understanding of the significance of the ERAD machinery, as well as its underlying mechanism, in disease pathogenesis remains limited.

Among many different ERAD machineries, the suppressor of lin-12-like (SEL1L, Hrd3p in yeast)- HMG-CoA reductase degradation 1 (HRD1, Hrd1p in yeast) complex represents the most conserved branch of ERAD from yeast to humans[6–9]. In both yeast and mammals, Hrd3p/

SEL1L not only regulates the stability of Hrd1p/HRD1[10–12], but also is involved in substrate recruitment by interacting with lectins such as Os9p/OS9 (Osteosarcoma amplified 9) and ERLEC1 (ER lectin 1)[13–15]. Biochemical and genetic studies in yeast and mammalian cells have shown that misfolded proteins in the ER are recruited to HRD1 through Yos9/Os9 and Hrd3p/SEL1L[15–17], retrotranslocated and ubiquitinated through the activities of Hrd1p/HRD1-Der1p/DERLIN retrotranslocon and the E2 Ubc6p/UBE2J1[18–20], and subsequently exacted from the ER membrane by the cytosolic AAA-ATPase Cdc48/VCP[21–23] for proteasomal degradation. In addition, human HERP (Usa1p in yeast) promotes HRD1 oligomerization as well as the formation of ERAD complex in vitro[24–27]. However, how the HRD1 protein complex is assembled together in vivo remains largely mysterious.

[1]Department of Molecular Physiology and Biological Physics, University of Virginia, School of Medicine, Charlottesville, VA 22903, USA. [2]Department of Molecular & Integrative Physiology, University of Michigan Medical School, Ann Arbor, MI 48105, USA. [3]Zilkha Neurogenetic Institute, Keck School of Medicine of University of Southern California, Los Angeles, CA 90033, USA. [4]Department of Pharmacology, University of Virginia, School of Medicine, Charlottesville, VA 22908, USA. [5]Present address: Life Sciences Institute and Department of Cell & Developmental Biology, University of Michigan Medical School, Ann Arbor, MI 48109, USA. [6]Present address: Department of Radiology and Biomedical Imaging, University of California San Francisco, San Francisco, CA 94143, USA. ✉e-mail: bjk5fz@virginia.edu; xvr2hm@virginia.edu

In mammals, global or acute deletion of *Sel1L* or *Hrd1* in germline and adult mice leads to embryonic or premature lethality, highlighting the crucial roles of SEL1L and HRD1 in both embryonic development and adult stages[10,28–30]. Recent studies using cell type-specific mouse models have further established a vital importance of SEL1L and HRD1 in a cell type- and substrate-specific manner in physiological process[3,31–54]. However, the role and importance of SEL1L in HRD1 ERAD remains uncertain. Indeed, studies have questioned the necessity of SEL1L in HRD1 ERAD: (1) Hrd3p/SEL1L function can be bypassed by overexpressing Hrd1p/HRD1 in both yeast and mammalian cell culture systems[11,55,56]; (2) In yeast, Hrd3p is dispensable for the interaction of Hrd1 with substrates and E2 enzyme, although it is required for the E3 ubiquitination activity of Hrd1 (or substrate ubiquitination)[12]; and (3) in mammals, additional SEL1L-independent HRD1 complexes have been proposed or identified, such as the HRD1-FAM8A1 complex[57–59].

We recently reported the identification 4 SEL1L and HRD1 variants in 11 patients with ERAD-associated neurodevelopmental disorders with onset infancy (EDNI) syndrome[60,61]. All these variants impair ERAD function via distinct mechanisms such as SEL1L-HRD1 complex stability, substrate recruitment and ubiquitination[60,61]. In analyzing another *SEL1L*[S658P] variant (*p.Ser658Pro*; c.1972T>C, NM_005065), first reported in 2012 in Finnish Hound suffering cerebellar ataxia with uncertain causality[62], we show that this SEL1L variant causes ERAD dysfunction, leading to developmental delay and early onset, non-progressive, cerebellar ataxia in mice. Proteomic screens of SEL1L and HRD1 interactomes reveal that SEL1L is required for the formation of an HRD1 functional complex by recruiting not only lectins OS9 and ERLEC1, but also the E2 enzyme UBE2J1 and DERLIN to HRD1.

## Results

### Generation of knock-in (KI) mice carrying *SEL1L*[S658P] variant

To establish disease causality of the *SEL1L*[S658P] variant, we generated knock-in (KI) mice carrying this variant on the B6SJLF1/J mix genetic background using the CRISPR/Cas9-based methods (Supplementary Fig. 1a). Two independent founders were established by PCR and sequencing (Supplementary Fig. 1b, c) and then used to generate bi-allelic homozygous KI mice by intercrossing the F1 generations of a cross between the founders and C57BL/6J WT mice. Each founder line was characterized independently. As similar results were obtained from each founder line, the results were combined.

Much to our surprise, at postnatal day 21, we obtained homozygous KI pups at a frequency of less than 11% in 247 pups of over 30 litters from 12 breeding pairs, less than half of the expected Mendelian transmission of a recessive trait (25%) (Supplementary Fig. 1d). Timed pregnancy followed by PCR-genotyping of the embryos at embryonic day (E) 10.5–12.5 during organogenesis and at E14.5–16.5 during body-mass growth and organ maturation showed that empty deciduae (i.e. reabsorbed embryos) were found at E10.5 and all turned out to be KI embryos (Fig. 1a). Indeed, the frequency of KI embryos in a total of over 100 embryos was progressively reduced from 17% at E10.5–12.5 to 14% at E14.5–16.5, while the frequency of heterozygous mice (HET) maintained at ~58% (Supplementary Fig. 1d). Hence, these data demonstrate that the *SEL1L*[S658P] allele is a recessive mutation and causes partial embryonic lethality at ~50% frequency in this mixed genetic background.

### Mild developmental delay of *SEL1L*[S658P] KI mice

In the surviving cohorts, KI mice exhibited modest growth retardation by ~10–15% in both sexes, starting at 3-4 weeks through the first 40 weeks of age (Fig. 1b). At 5 weeks of age, the KI mice of both sexes were slightly shorter in body length compared to that of WT litter-mates (Fig. 1c). This growth delay was not due to reduced food intake or associated with abnormal blood glucose levels (Fig. 1d and Supplementary Fig. 1e). Blind histological examination of peripheral tissues of 5-week-old KI mice by a pathologist revealed no obvious

histological abnormalities in the pancreas, liver, kidneys, inguinal white adipose tissue (iWAT), and brown adipose tissue (BAT) (Fig. 1e, f and Supplementary Fig. 1f). Biochemical analyses of serum revealed no significant liver damage in both sexes based on the levels of alanine transaminase (ALT), alkaline phosphatase (ALP) and total bilirubin (TBIL) (Fig. 1g), or in the kidneys based on blood urea nitrogen (BUN) and creatinine levels (CREA) (Fig. 1h). Of note, although ALT level was significantly increased in male KI mice compared to that of WT litter-mates, it was still within the normal range (24.3–115.3 U/L) (Fig. 1g). Hence, *SEL1L*[S658P] is a recessive hypomorphic mutation causing mild growth retardation, while having no major impact on peripheral tissues in mice.

### Early-onset, non-progressive, cerebellar ataxia in *SEL1L*[S658P] KI mice

KI mice developed abnormal hind limb-clasping reflexes when suspended by their tails, starting at 3–9 weeks of age (Fig. 2a, b). Intriguingly, the phenotype remained mild for up to 48 weeks of age, pointing to the non-progressive nature of the disease (Fig. 2a, b). Unlike WT mice showing a normal symmetric gait characterized by the overlapping of hindlimb and forelimb paw prints, KI mice exhibited asymmetric gait patterns characterized by the lagging of the hindlimbs and reduction in stride length at 6 weeks of age (arrows, Fig. 2c, quantitated in d). Interestingly, while the balance beam test showed that KI mice at 6 and 12 weeks of age had no problems completing the test compared to WT littermates (Fig. 2e, f), some KI mice slipped and lost their balance at least once while crossing the 1-meter beam (Fig. 2e, g and Supplementary Movie 1), pointing to an impairment of balance. Moreover, KI mice showed worse performance of motor coordination in the rotarod tests compared to WT littermates even after 3 days of training at 6 weeks of age (Fig. 2h, i). HET mice were comparable to WT littermates in hindlimb clasping, gait pattern and rotarod tests (Fig. 2a–d, h, i), in line with the autosomal recessive nature of the variant. These data show that *SEL1L*[S658P] KI mice exhibit mild growth retardation, and signs of early onset non-progressive mild ataxia, establishing the disease-causality and pathogenicity of this allele.

### Microcephaly and a mild reduction of Purkinje cells in *SEL1L*[S658P] KI mice

We next explored the molecular changes in the brain that may underlie the pathogenesis of early-onset ataxia. At 5 weeks of age, KI mice had reduced brain weight by ~10% compared to those of age-matched WT and HET littermates (Fig. 3a). MRI analyses revealed smaller cere-bellum and cortex in KI mice by ~10% relative to those of WT litter-mates, i.e. microcephaly (Fig. 3b and Supplementary Fig. 2a). Blind histological examination by a pathologist revealed no obvious histo-logical abnormalities in various brain regions, including the cortex, of KI mice at 5 weeks of age (Fig. 3c, d). Immunostaining of markers for neurons (NeuN), astrocytes (GFAP), and microglia (Iba1) also showed comparable staining pattens in the whole brain and the cortex of KI vs. WT mice at 5 weeks of age (Fig. 3e and Supplementary Fig. 2b).

Given the cerebellar ataxia phenotype of both the affected canine[62] and KI mice (Fig. 2), we next examined the cerebellar region consisted of Purkinje neurons. Purkinje cells are arranged in a single layer in the cerebellar cortex, with their dendrites facing the molecular layer (ML) and their axons facing the white matter (WM) (Fig. 3f, g). Alterations in Purkinje cell function have been directly associated with cerebellar ataxia[63–65]. Quantitation of the number of Purkinje cells based on their unique location and morphology in H&E-stained sagittal sections of the cerebellum revealed a mild 15–20% reduction in KI mice at 5 weeks of age (Fig. 3f, h). This reduction did not worsen with age as a similar level of reduction was observed at 24 weeks of age (Fig. 3g, h). This result was further confirmed by the quantitation of the Purkinje cell marker Calbindin using Western blot (Fig. 3i) and immunostaining at both 5 and 24 weeks of age (Supplementary Fig. 2c). In addition,

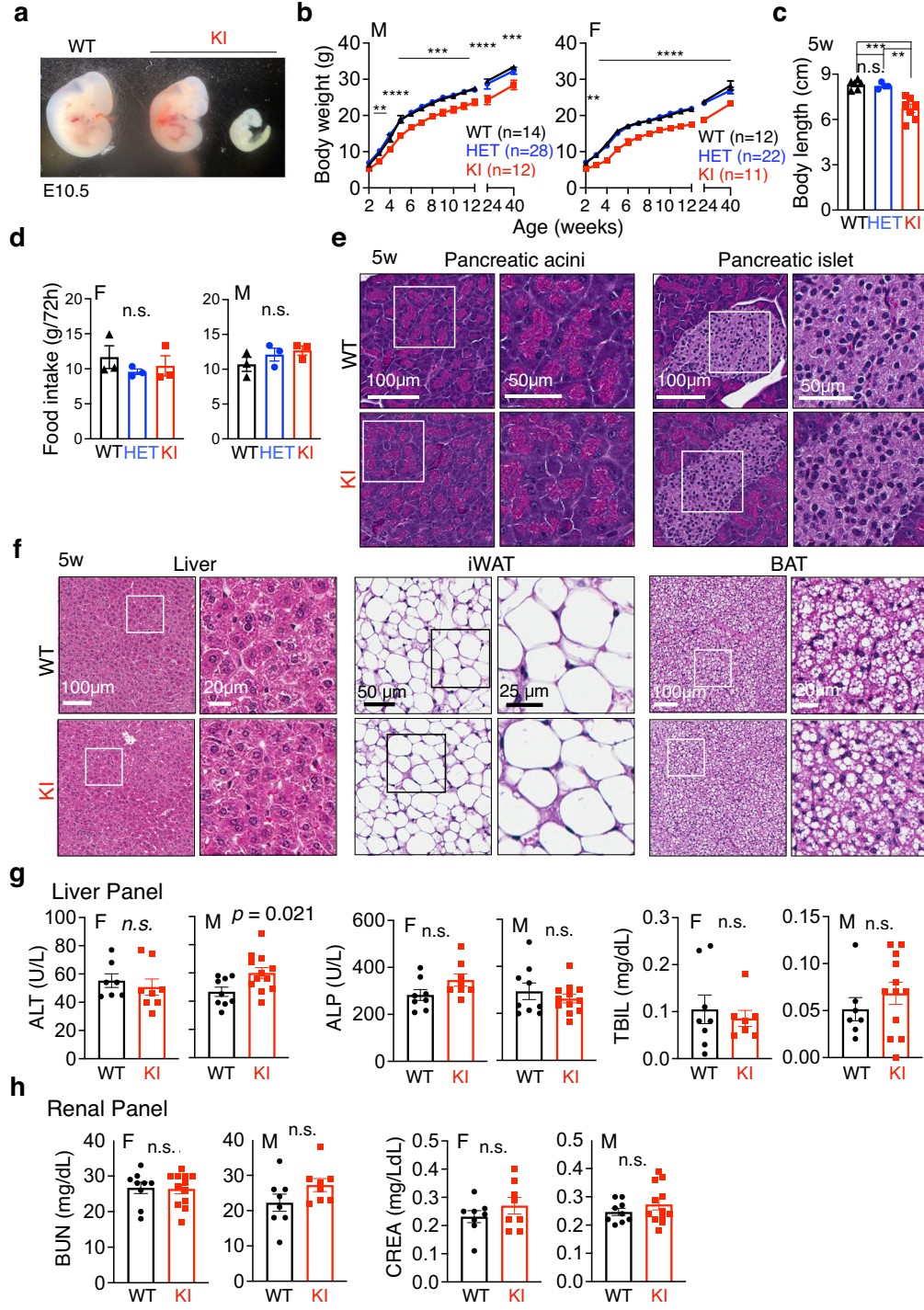

**Fig. 1 | *SEL1L^{S658P}* KI mice exhibit partial embryonic lethality and developmental delay. a** Gross appearance of *SEL1L^{S658P}* KI embryos (one dead, right) compared with their WT littermates at E10.5. **b** Growth curve. $n = 12$, 28 and 12 mice for male WT, HET and KI; $n = 12$, 22 and 11 mice for female WT, HET and KI. **c** Body length of 5-week-old mice of both sexes ($n = 6$, 4 and 7 mice for WT, HET and KI). **d** 72 h food intake of 5-week-old mice of both sexes ($n = 3$ mice per group). **e**, **f** Hematoxylin & eosin (H&E) images of various peripheral tissues (**e**, pancreatic acini and islet; **f**, Liver, iWAT and BAT) from 5-week-old mice ($n = 3$ mice per group). iWAT, inguinal white adipose tissue; BAT, brown adipose tissue. **g** Serum levels of alanine transaminase (ALT), total bilirubin (TBIL) and alkaline phosphatase (ALP) in

5-week-old mice of both sexes (ALT: $n = 7$, 8, 9 and 12 mice for female WT, KI, male WT and KI; ALP: $n = 8$, 8, 9 and 12 mice for female WT, KI, male WT and KI; TBIL: $n = 8$, 7, 7 and 12 mice for female WT, KI, male WT and KI). **h** Serum levels of blood urea nitrogen (BUN) and creatinine (CREA) in 5-week-old mice of both sexes (BUN: $n = 9$, 12, 8 and 8 mice for female WT, KI, male WT and KI; CREA: $n = 8$, 8, 9 and 12 for female WT, KI, male WT and KI). Values, mean ± SEM. n.s. not significant; **$p < 0.01$, ***$p < 0.001$ and ****$p < 0.0001$ by two-way ANOVA followed by Tukey's multiple comparisons test (**b**), one-way ANOVA followed by Tukey's post hoc test (**c**, **d**) and two-tailed Student's *t* test (**g**, **h**).

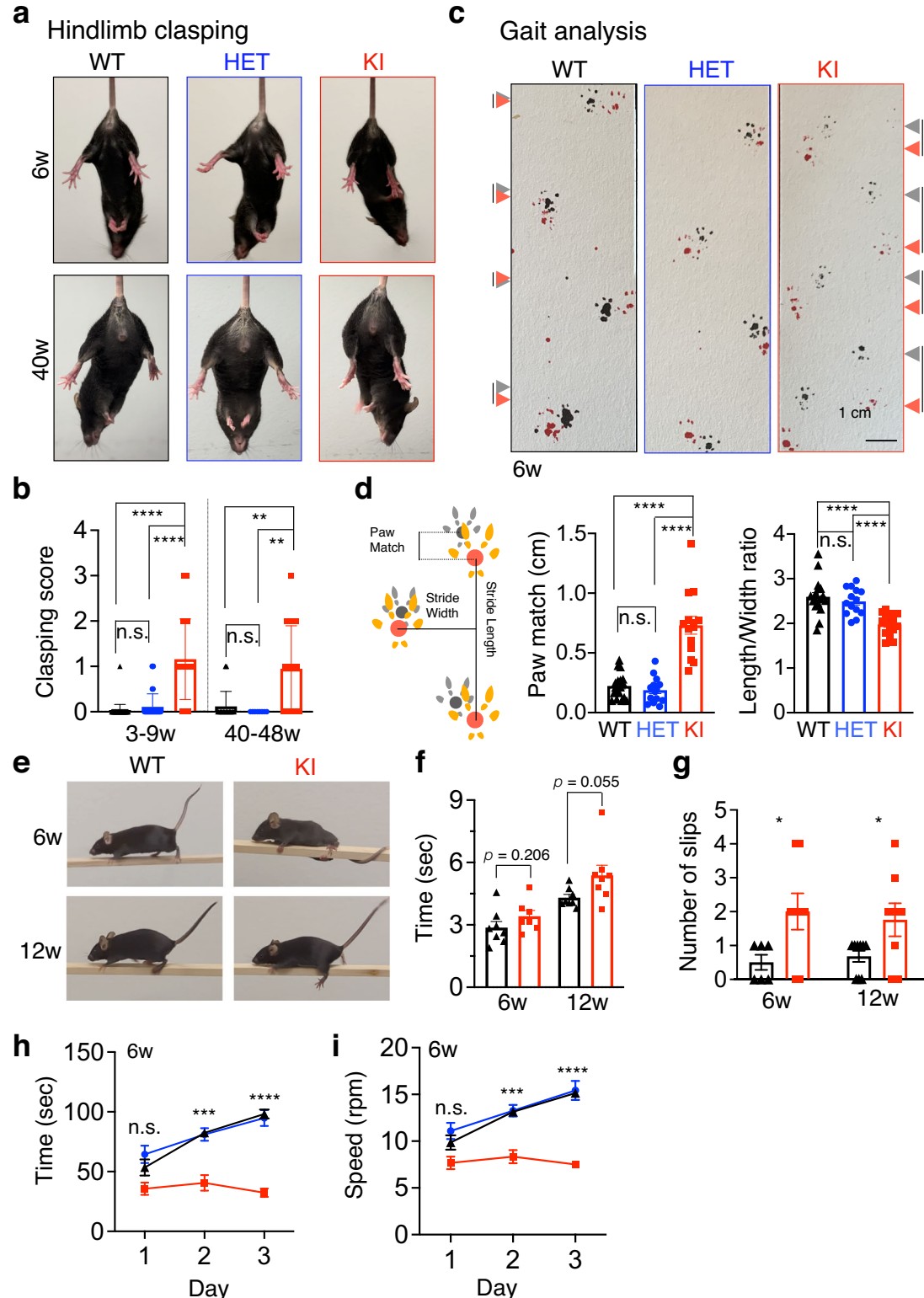

**Fig. 2 | *SEL1L^S658P* KI mice exhibit early-onset, non-progressive, ataxia. a, b** Representative pictures of hindlimb clasping (**a**) and score quantitation (**b**) of 3–9 ($n = 49$, 51 and 38 for WT, HET and KI) and 40-48 ($n = 17$, 8 and 20 mice for WT, HET and KI) -week-old littermates of both sexes. **c, d** Representative pictures (**c**), cartoon schematic of paw prints (left, **d**) and quantitation of gait analysis (right, **d**) of 6-week-old littermates of both sexes. The gray and red arrows in (**c**) indicate forelimb and hindlimb, respectively; and the lines between gray and red arrow indicate the distance between the two limbs ($n = 18$, 14 and 14 mice for WT, HET and KI). **e, f, g** Representative pictures of slips on balance beam (**e**) of 6 and 12-week-old KI mice, with quantitation of time of beam crossing (**f**, $n = 8$, 7, 7 and 7 mice for 6w WT, KI, 12w WT and KI) and the number of slips shown in (**g**, $n = 6$, 8, 9 and 8 mice for 6w WT, KI, 12w WT and KI). **h, i** Quantitation of time (**h**) and rpm (**i**) of rotarod test from 6-week-old mice with 3 days training ($n = 7$, 12 and 6 mice for WT, HET and KI). Values, mean ± SEM. n.s., not significant; *$p < 0.05$, **$p < 0.01$, ***$p < 0.001$ and ****$p < 0.0001$ by one-way ANOVA followed by Tukey's post hoc test (**b, d**), two-tailed Student's *t* test (**f, g**) and two-way ANOVA followed by Tukey's multiple comparisons test (**h, i**).

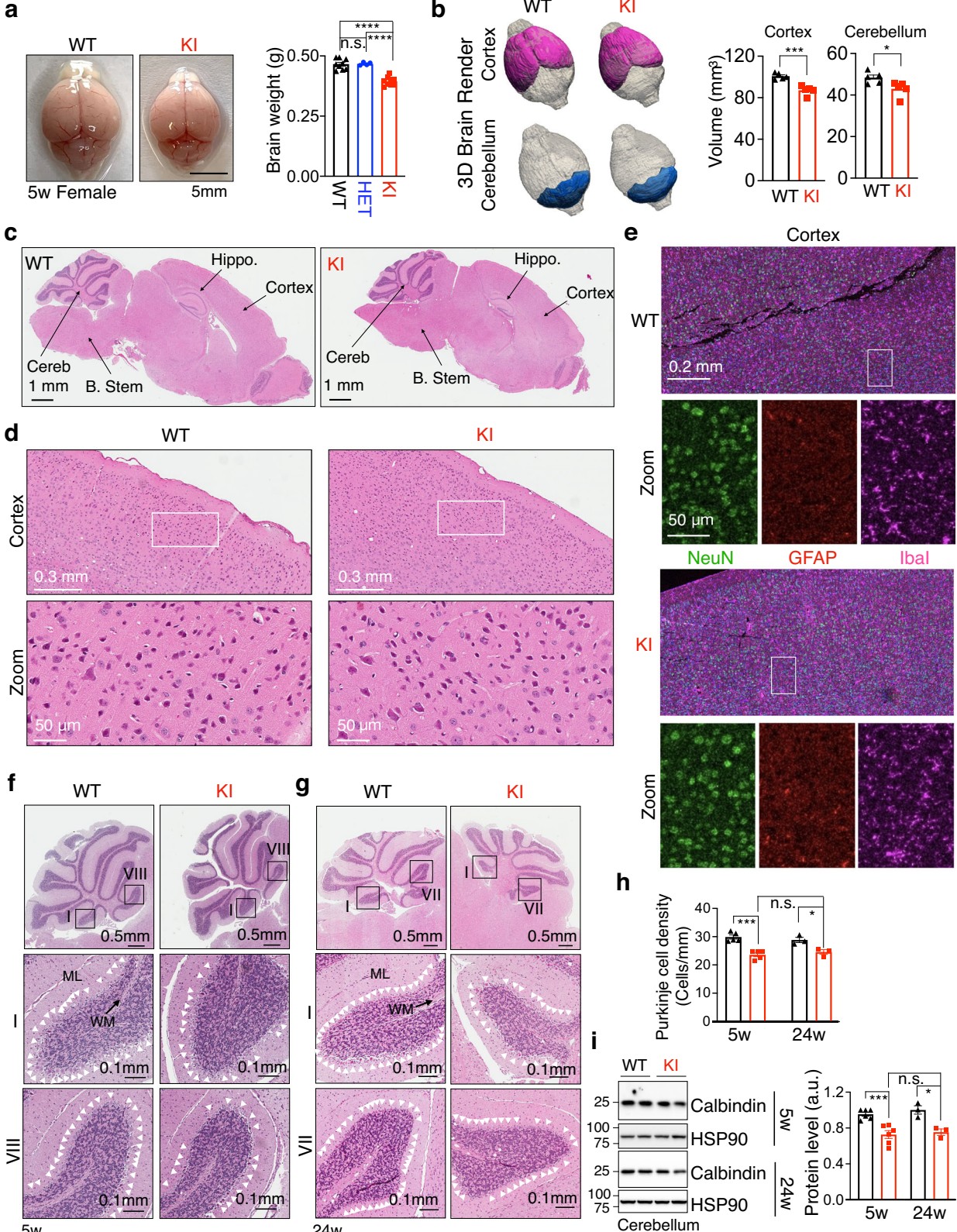

**Fig. 3 | *SEL1L^{S658P}* KI mice exhibit microcephaly and a mild reduction of Purkinje cells. a** Representative pictures of brains of 5-week-old mice, with quantitation of brain weight shown on the right ($n = 9$, 4 and 9 mice for WT, HET and KI). **b** 3D rendering of cortex and cerebellum from 5-week-old mice, with quantitation shown on the right ($n = 5$ mice per group). **c, d** Hematoxylin & eosin (H&E) stained sagittal sections of whole brain (**c**) and cortex (**d**) from 5-week-old mice ($n = 3$ mice per group). Hippo., hippocampus; B. Stem, brain stem; Cereb., cerebellum. **e** Representative confocal images of NeuN (green), GFAP (red) and IbaI (purple) staining for neurons, astrocytes and microglia, respectively, in the cortex of 5-week-

old mice ($n = 2$ mice per group). **f, g, h** Hematoxylin & eosin (H&E) stained sections of the cerebellum from 5- (**f**) and 24- (**g**) week-old mice, with quantitation of Purkinje cell density shown in (**h**) ($n = 5$ mice for 5w WT and KI; $n = 3$ for 24w WT and KI). White arrowheads, Purkinje cells. ML, molecular layer; MW, white matter. **i** Western blot analysis of Calbindin in the cerebellum of 5- and 24-week-old mice with quantitation shown on the right ($n = 6$ mice for 5w WT and KI; n = 3 for 24w WT and KI). Values, mean ± SEM. n.s., not significant; *$p < 0.05$, ***$p < 0.001$ and ****$p < 0.0001$ by one-way ANOVA followed by Tukey's post hoc test (**a**) and two-tailed Student's *t* test (**b, h, i**).

examination of other neurons, astrocytes and microglia in the cerebellum of KI mice at 5 weeks of age using immunostaining and Western blot revealed no obvious changes in these populations compared to WT littermates (Supplementary Fig. 3a, b). Hence, our data show that *SEL1L*[S658P] is associated with microcephaly and a mild, but non-progressive, reduction of Purkinje cells starting at a young age.

### Cellular adaptation in the cerebellum of KI mice

As ER stress was implicated in *SEL1L*[S658P] canine[62], we next asked whether and how ER homeostasis was affected in the cerebellum of *SEL1L*[S658P] KI mice. In line with our previous study that UPR sensor IRE1α is an ERAD substrate[33], its protein level was increased by ~4 folds; however, neither IRE1α phosphorylation nor splicing of its downstream effector *Xbp1* mRNA was elevated in the cerebellum of 5-week-old KI mice (Fig. 4a). While phosphorylation of another UPR sensor PERK and its downstream effector eIF2α was enhanced by 1.5-2 folds, quantitation of the ratio of phosphorylated to total proteins showed a subtle activation of UPR (Fig. 4b). Expression of CHOP, a downstream mediator of UPR, was mildly elevated by 2 folds in the cerebellum of 5-week-old KI mice compared to WT littermates (Fig. 4c). Moreover, protein levels of key ER chaperones such as BiP, GRP94 and protein disulfide isomerase (PDI) were modestly increased by 20–40 and 100%, respectively, in the cerebellum of KI mice relative to those in WT littermates (Fig. 4d). In keeping with the subtle changes of UPR, transmission electron microscopic (TEM) examination of Purkinje cells revealed largely normal ER sheet morphology in both 3- and 24-week-old KI mice (Fig. 4e, f). Cleaved caspase 3, a marker of apoptosis, was undetectable in the cerebellum of KI mice at either 5 or 24 weeks of age (Fig. 4g). Hence, these data suggest that neurons including Purkinje cells can adapt to the expression of *SEL1L*[S658P] without eliciting an overt ER stress or cell death. This conclusion was further supported by the largely normal histology and function of peripheral tissues such as the liver, kidneys and pancreas (Fig. 1).

### *SEL1L*[S658P] is a hypomorphic variant with impaired ERAD function

We next explored whether and how *SEL1L*[S658P] causes HRD1 ERAD dysfunction both in vivo and in vitro. Confocal microscopic analysis revealed that SEL1L is highly enriched in the ER of Purkinje cells in the cerebellum (Fig. 5a and Supplementary Fig. 4). Western blot analysis showed that SEL1L and HRD1 protein levels were reduced by 20 and 60%, respectively, in the cerebellum of 5-week-old KI mice relative to those in WT littermates (Fig. 5b, c). These changes were associated with the accumulation of two known ERAD substrates IRE1α[33] (Fig. 4a) and CD147[66] (Fig. 5b, c).

In vitro, in knock-in (KI) HEK293T cells carrying the bi-allelic *SEL1L*[S658P] variant (Supplementary Fig. 5a, b), protein levels of SEL1L and HRD1 were reduced by 20 and 30%, respectively, compared to WT HEK293T cells (Fig. 5d, e). Indeed, *SEL1L*[S658P] protein was unstable and, to a lesser extent, for HRD1 (Fig. 5f, g). Nonetheless, *SEL1L*[S658P] expression led to a significant accumulation, as a result of protein stabilization, of several known ERAD substrates such as IRE1α[33], OS9[67], and CD147[66] in KI cells compared to those in WT HEK293T cells, but to a much less extent when compared to those in *SEL1L*[−/−] HEK293T cells (Fig. 5d–g). In line with these findings, turnover of a known ERAD substrate, a disease mutant of pro-arginine vasopressin (proAVP) at residue 57 (Gly-to-Ser, Gly57Ser)[36], was attenuated in *SEL1L*[S658P] transfected *SEL1L*[−/−] HEK293T cells, leading to its accumulation and the formation of HMW proAVP aggregates (lane 6–8 vs. 3–5, Fig. 5h and quantitated in Fig. 5i). Hence, these data demonstrate that *SEL1L*[S658P] impairs HRD1 ERAD function while having a mild impact on SEL1L-HRD1 protein stability.

### Sequence and structural analyses of *SEL1L*[S658P] variant

We next sought to explore additional mechanism(s) underlying the impact of the variant on ERAD function using the sequence-structural

analyses. SEL1L S658 is located at the Sel1-like repeat-C (SLR-C) domain, and highly conserved from Drosophila to humans, but not in yeast (Fig. 6a, b). Position-specific scoring matrix (PSSM) scores for residue 658 were positive for Ser (green) but negative for Pro (red, Supplementary Fig. 6a), suggesting that Ser has been evolutionarily selected and that Pro at this position might have detrimental effects on SEL1L function. AI-based structural prediction AlphaFold2 analysis for the human SEL1L (1-722 aa)- HRD1 (1-334 aa)-DERLIN-OS9 protein complex (Fig. 6c), based on the Cryo-EM structure of the yeast protein complex[18] (Supplementary Fig. 6b) showed that the structure for the human complex was likely quite similar to that of the yeast (Supplementary Fig. 6b). Notably, S658 is located on an α-helix in a close proximity to the amphipathic helix of SEL1L[18,68], which may directly interact with the transmembrane 1 (TM1)–TM2 loop of HRD1 (Fig. 6c, d). In addition, SEL1L S658 is in proximity to residues E659 and R655 of SEL1L, which may be involved in the interaction with OS9 (Supplementary Fig. 6c).

### *SEL1L*[S658P] variant attenuates its interaction with HRD1, but not OS9

We next asked whether *SEL1L*[S658P] interferes with the formation of ERAD complex. In *SEL1L*[−/−] HEK293T cells transfected with FLAG-tagged WT or S658P SEL1L, SEL1L S658P markedly attenuated the interaction between SEL1L and endogenous HRD1 compared to that of SEL1L WT, while having no effect on its interaction with other components of the complex including OS9 and ERLEC1 (Fig. 6e). Similar results were obtained in *SEL1L*[−/−] HEK293T cells overexpressing both SEL1L-FLAG and HRD1-Myc (Supplementary Fig. 6d). Furthermore, mutation of Ser658 to Ala (S658A) had no effect on the SEL1L-HRD1 interaction (lane 4 vs. 2–3, Fig. 6f), while S658D or K attenuated the interaction, albeit not as dramatically as S658P (lane 5–6 vs. 2–3, Fig. 6f). None of these mutations at SEL1L S658 affected its interaction with OS9 (Fig. 6f). Mutating P658 back to S (P658S) rescued the interaction between SEL1L and HRD1 (lane 7, Fig. 6f), excluding the possibility of additional mutations outside of S658P. In addition, mutation of a nearby Ser at position 663, 5 amino acids downstream of S658 located at the loop between the two α-helices (Fig. 6g), to Pro (S663P) had no apparent effect on SEL1L-HRD1 interaction (lane 8, Fig. 6f). In line with the overexpression experiments, SEL1L interaction with HRD1 was reduced by 5 folds in KI livers, while having no effect on the interactions with OS9 and ERLEC1, compared to those in WT and HET livers (Fig. 6h). Hence, we conclude that *SEL1L*[S658P] attenuates SEL1L-HRD1 interaction while having no effect on the SEL1L-lectins interactions.

### *SEL1L*[S658P] attenuates the interaction with HRD1 via SEL1L F668 - HRD1 Y30

We then investigated mechanistically how SEL1L S658P attenuates SEL1L-HRD1 interaction. Structural predication revealed a potential physical interaction between SEL1L-F668 (on a neighboring helix) and HRD1-Y30 (on TM1 of HRD1) through an aromatic-aromatic interaction in an "edge-to-face" attractive orientation (Fig. 7a, b). However, in *SEL1L*[S658P], F668-Y30 interaction would be predictably changed to a "face-to-face" orientation (Fig. 7b), which may result in an unfavorable electrostatic repulsion force between the two planar faces of aromatic rings[69].

Both SEL1L-F668 and HRD1-Y30 residues are highly conserved from Drosophila to humans, but not in yeast (Supplementary Fig. 7a, b). We next examined whether SEL1L F668 or HRD1 Y30 is critical for the SEL1L-HRD1 interaction. Indeed, mutations of HRD1 Y30 to Ala (A), Asp (D) or Lys (K) significantly disrupted the SEL1L-HRD1 interaction by 80-90%, and 30% when mutated to Phe (F) (Supplementary Fig. 7c). The disruption of SEL1L-HRD1 interaction abolished the interaction of HRD1 with OS9, while having no impact on HRD1 interaction with FAM8A1 (Supplementary Fig. 7c). We then mutated F668 to Ala (F668A), which disrupted the SEL1L-HRD1 interaction similarly to

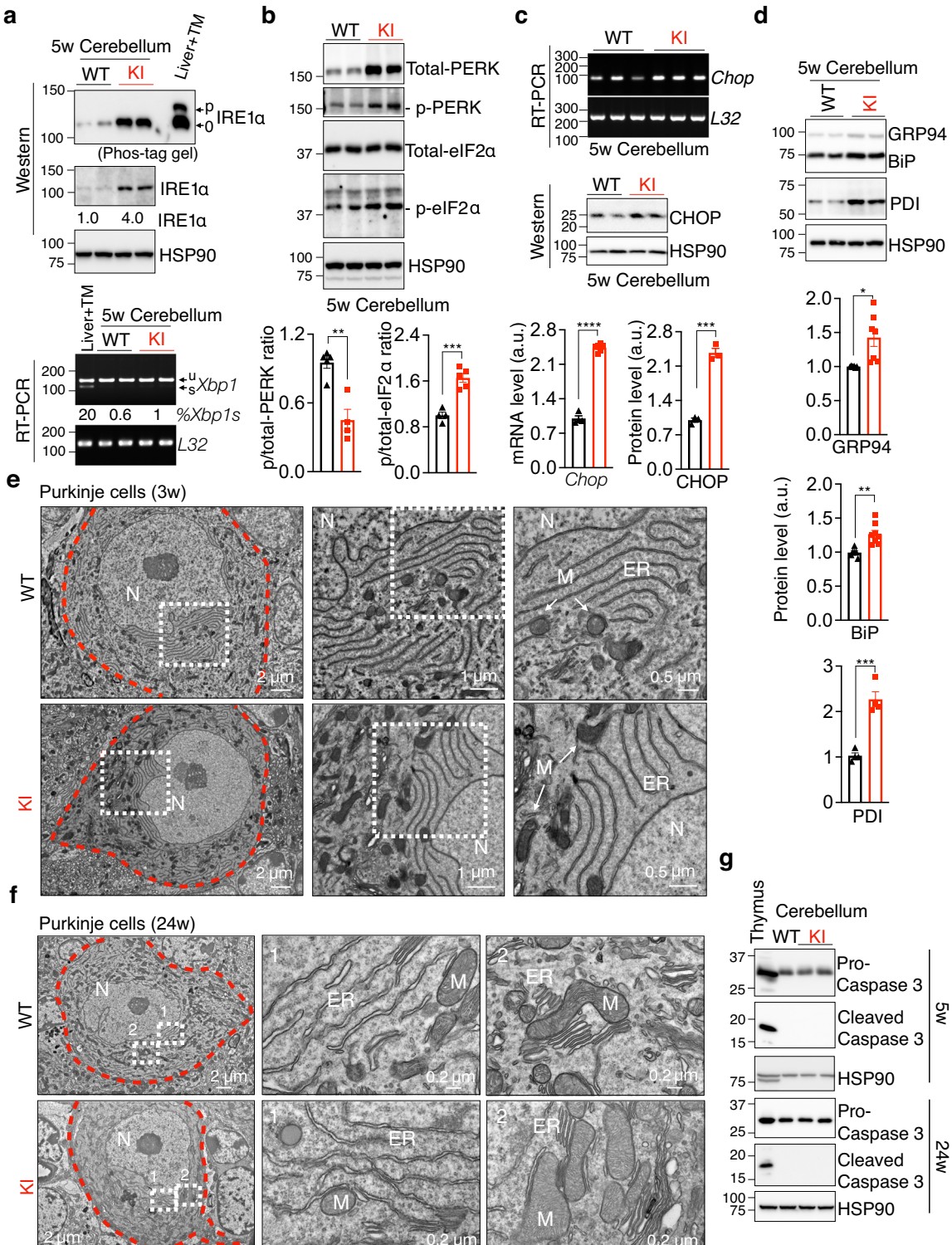

**Fig. 4 | Subtle alteration of ER homeostasis in the cerebellum of *SEL1L$^{S658P}$* KI mice. a** Phos-tag-based Western blot analysis of IRE1α phosphorylation (top) and RT-PCR of *Xbp1* splicing in the cerebellum of 5-week-old mice. 0/p, non-/phosphorylation; u/s, un-/spliced *Xbp1*. Quantitation of IRE1α and the ratio of spliced to total *Xbp1* shown below the gel as mean (*n* = 4-5 mice per group). Livers from mice injected with tunicamycin (TM, an ER stress inducer) included as a positive control. **b** Western blot analysis of PERK and eIF2α phosphorylation in the cerebellum of 5-week-old mice, with quantitation of the ratio of phosphorylated to total PERK (*n* = 5, 4 mice for WT and KI) or eIF2α (*n* = 4, 5 mice for WT and KI) shown below. **c** RT-PCR and Western blot analyses of CHOP mRNA (*n* = 4, 6 mice for WT and KI) and protein

levels (*n* = 3 mice for WT and KI) in the cerebellum of 5-week-old mice, with quantitation shown below. **d** Western blot analysis of ER chaperones GRP94, BiP and PDI in the cerebellum of 5-week-old mice with quantitation shown below (GRP94/BiP: *n* = 6, 7 mice for WT and KI; PDI: *n* = 4 mice for WT and KI). **e, f** Representative TEM images of Purkinje cells (outlined by red dotted lines) of 3- (**e**) and 24- (**f**) week-old mice. N, nucleus; M, mitochondria. *n* = 2 mice each genotype at each age. **g** Western blot analysis of pro- and cleaved Caspase 3 in the cerebellum of 5- and 24-week-old mice. Thymus, a positive control. *n* = 3 mice each genotype. Values, mean ± SEM. *$p < 0.05$, **$p < 0.01$, ***$p < 0.001$ and ****$P < 0.0001$ by two-tailed Student's *t* test (**b**–**d**).

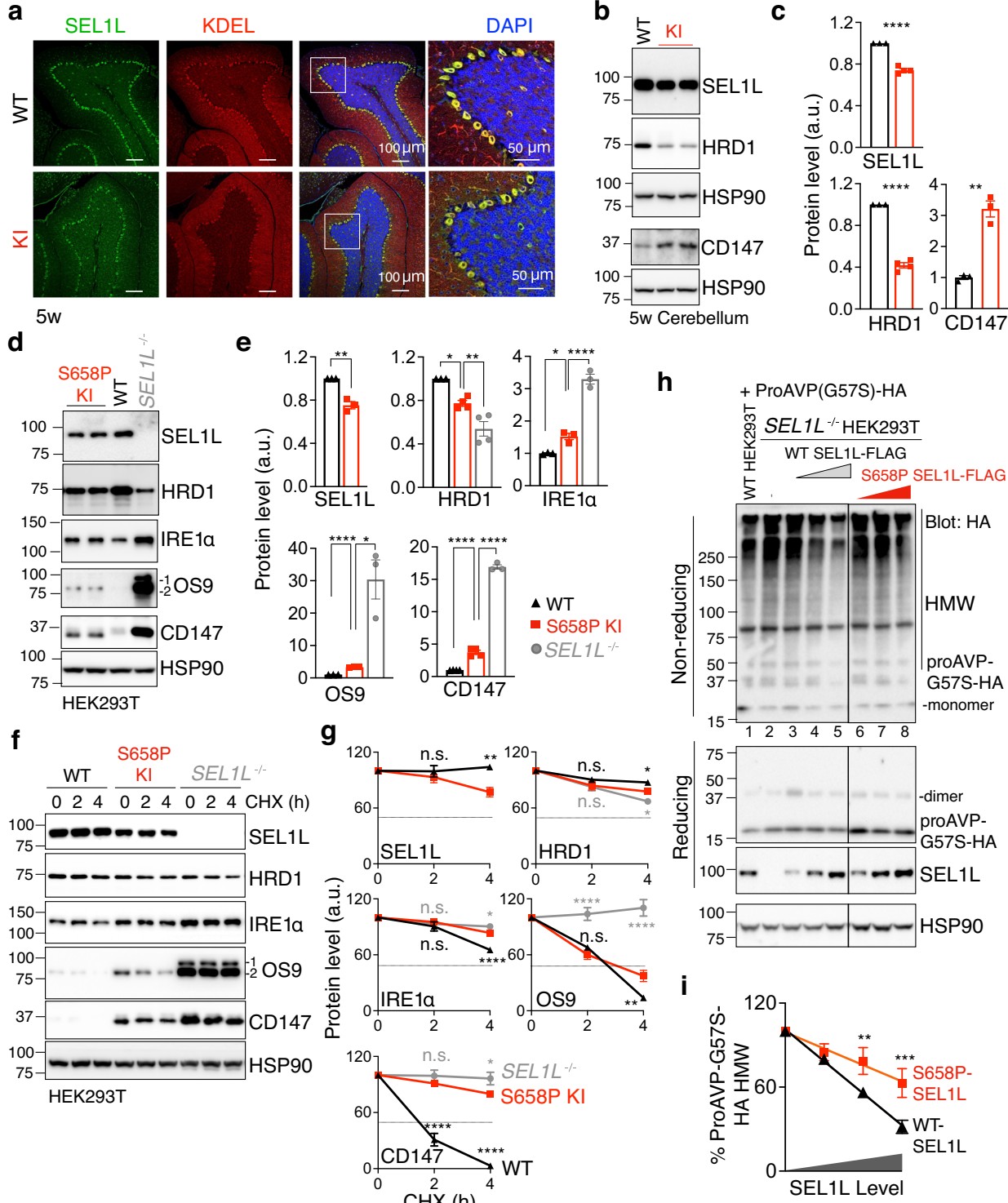

**Fig. 5 | *SEL1L^{S658P}* impairs SEL1L-HRD1 protein stability and causes ERAD dysfunction. a** Representative confocal images of SEL1L (green) and KDEL (ER marker, red) in the cerebellum of 5-week-old mice (two independent repeats). **b, c** Western blot analysis of SEL1L, HRD1 and known endogenous substrate CD147 in the cerebellum of 5-week-old mice, with quantitation shown in (**c**) (SEL1L/HRD1: $n = 3$, 4 mice for WT and KI; CD147: $n = 3$ mice for WT and KI). **d, e** Western blot analysis of SEL1L, HRD1 and known endogenous substrates IRE1α, OS9 and CD147 in WT, *SEL1L^{S658P}* KI or *SEL1L^{-/-}* HEK293T cells, with quantitation shown in (**e**) (SEL1L/IRE1α/OS9: $n = 3$ each genotype; HRD1: $n = 3$, 4 and 4 for WT, KI and *SEL1L^{-/-}*; CD147: $n = 4$, 4 and 3 for WT, KI and *SEL1L^{-/-}*; "n" indicates independent samples). **f, g** Cycloheximide (CHX) chase analysis of SEL1L, HRD1 and known endogenous substrates IRE1α, OS9 and CD147 in WT, *SEL1L^{S658P}* KI or *SEL1L^{-/-}* HEK293T cells, with quantitation shown in (**g**)

(SEL1L: $n = 3$, 4 for WT and KI; HRD1/ IRE1α/OS9: $n = 3$, 4 and 3 for WT, KI and *SEL1L^{-/-}*; CD147: $n = 5$ each genotype; "n" indicates independent samples). **h, i** Reducing and non-reducing SDS-PAGE and Western blot analyses of proAVP-G57S high molecular-weight (HMW) aggregates in WT or *SEL1L^{-/-}* HEK293T cells transfected with indicated SEL1L-FLAG constructs in a dose-dependent manner, with quantitation of proAVP-G57S-HA HMW shown in (**i**) ($n = 7$ and 4 independent samples for WT and S658P SEL1L-FLAG). Two panels in (**h**) were from the same experiment with the irrelevant lanes in the middle cut off. Values, mean ± SEM. n.s., not significant; *$p < 0.05$, **$p < 0.01$, ***$p < 0.001$ and ****$p < 0.0001$ by two-tailed Student's $t$ test (**c**, SEL1L in **e**), one-way ANOVA followed by Dunnett's multiple comparisons test (**e**). two-way ANOVA followed by Dunnett's multiple comparisons test (**g, i**).

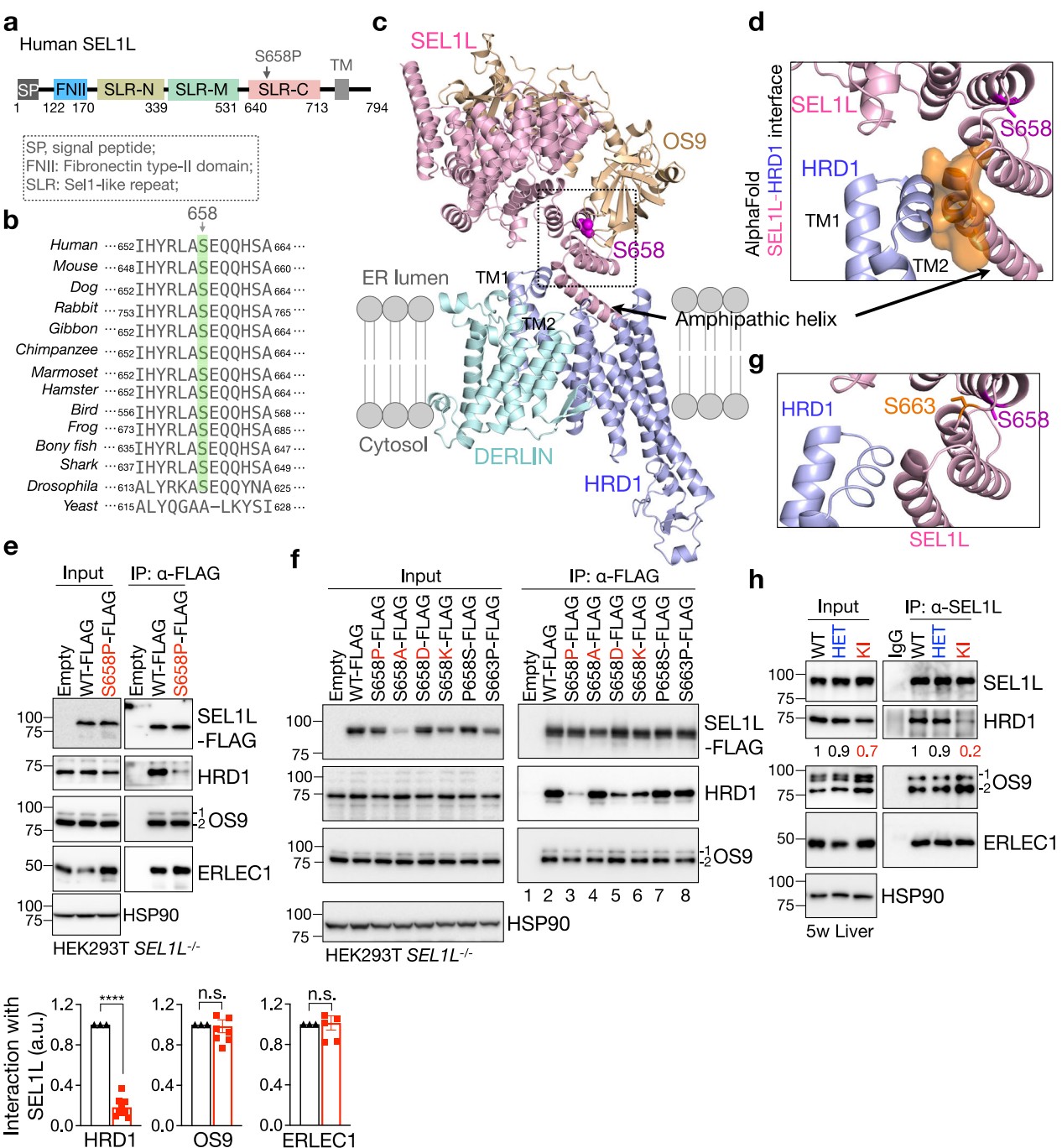

**Fig. 6 | SEL1L^S658P attenuates its interaction with HRD1. a** Schematic diagram of human SEL1L with domains and the location of the variant indicated. SP, signal peptide; FNII, fibronectin type II domain; SLR-N/M/C, Sel1-like repeats at N-, Middle- and C-terminal; TM, transmembrane. **b** Amino acid sequence alignment of SEL1L showing the conservation of SEL1L S658 residue across species (highlighted in green). **c** Structural prediction of human SEL1L/OS9/HRD1/DERLIN ERAD complex using AlphaFold2 with SEL1L S658 shown in magenta atoms. The arrow indicates the amphipathic helix of SEL1L that interacts with HRD1. **d** Side view of a space-filling model of the HRD1-SEL1L interface containing TM1-2 of HRD1 and the amphipathic helix of SEL1L. **e** Immunoprecipitation of FLAG-agarose in *SEL1L^−/−* HEK293T cells expressing indicated SEL1L-FLAG variants to examine their interaction with HRD1, OS9, and ERLEC1, with quantitation shown below (HRD1: $n = 9$ each genotype; OS9: $n = 8$ each genotype; ERLEC1: $n = 6$ each genotype; "n" indicates independent samples). Values, mean ± SEM. n.s. not significant; ****$p < 0.0001$ by two-tailed Student's *t* test. **f** Immunoprecipitation of FLAG-agarose in *SEL1L^−/−* HEK293T cells expressing indicated SEL1L-FLAG variants to examine their interaction with HRD1 and OS9 (two independent repeats). **g** Side view of the HRD1-SEL1L interface. SEL1L S658 and S663 residues are indicated in magenta and orange, respectively. **h** Immunoprecipitation of SEL1L in the livers from 5-week-old WT and KI mice to examine the interaction between endogenous SEL1L and HRD1, with quantitation of HRD1 shown below the blots as mean values (two independent repeats).

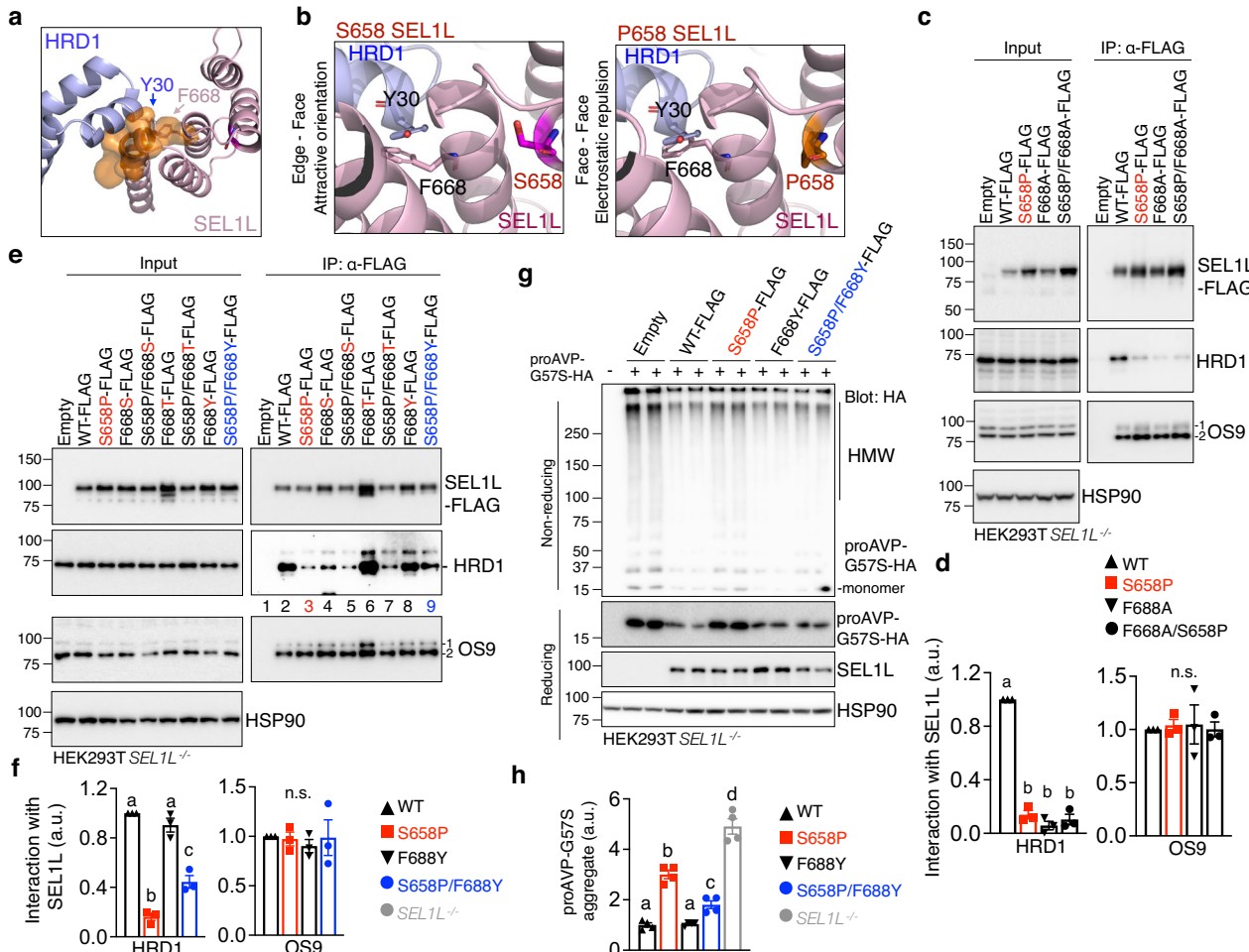

**Fig. 7 | SEL1L[S658P] disrupts its interaction with HRD1 via F668 (SEL1L)-Y30 (HRD1). a** Side view of SEL1L-HRD1 interaction interface to show HRD1-Y30 and SEL1L-F688 at the interface. **b** Side views of the aromatic-aromatic interaction between HRD1 Y30 and SEL1L F668 in SEL1L-WT (left) and S658P (right) variants. **c, d** Immunoprecipitation of FLAG-agarose in *SEL1L[−/−]* HEK293T cells expressing indicated SEL1L-FLAG variants to examine their interaction with HRD1 and OS9, with quantitation shown in (**d**) (*n* = 3 independent samples for each genotype). **e, f** Immunoprecipitation of FLAG-agarose in *SEL1L[−/−]* HEK293T cells expressing indicated SEL1L-FLAG variants to examine their interaction with HRD1 and OS9,

with quantitation shown in (**f**) (*n* = 3 independent samples for each genotype). **g, h** Reducing and non-reducing SDS-PAGE and Western blot analyses of proAVP-G57S high molecular-weight (HMW) aggregates in WT or *SEL1L[−/−]* HEK293T cells transfected with indicated SEL1L-FLAG constructs, with quantitation of proAVP-G57S-HA HMW shown in (**h**) (*n* = 4 independent samples for each genotype). Value, mean ± SEM. n.s. not significant; Different letters (abcd) indicate significant differences at the *p* = 0.05 level by one-way ANOVA followed by Tukey's post hoc test (**d, f, h**).

S658P but failed to rescue SEL1L S658P -HRD1 interaction (Fig. 7c, d). Similarly, mutations of F668 to other residues including Met (M), Asp (D), Lys (K), His (H), Ser (S), Cys (C) also disrupted the SEL1L-HRD1 interaction but had no effect on the SEL1L S658P -HRD1 interaction (Fig. 7e and Supplementary Fig. 7d, e). By contrast, mutations of F668 to Tyr (Y) or Trp (W), amino acids with an aromatic side chain, did not affect the interaction between SEL1L and HRD1 (Fig. 7e, f and Supplementary Fig. 7e). Similarly, mutation of F668 to Asn (N) with a carboxamide side chain had no effect on the SEL1L-HRD1 interaction (Supplementary Fig. 7e). Only F668Y partially rescued the SEL1L S658P -HRD1 interaction by 2 folds (lane 9 vs. 3, Fig. 7e, f). None of these SEL1L mutants affected the interaction between SEL1L and OS9 (Fig. 7c–f and Supplementary Fig. 7d, e).

We then asked whether the improved SEL1L-HRD1 interaction in SEL1L F668Y/S658P may improve HRD1 ERAD function. Indeed, SEL1L S658P/F668Y partially reversed the accumulation and HMW aggregation of proAVP-G57S protein upon transfected into *SEL1L[−/−]* HEK293T cells, albeit to a lesser extent than those transfected with SEL1L WT and F668Y (Fig. 7g, h). These data suggested that SEL1L S658P may cause a physical collision between SEL1L F668 and HRD1 Y30 residues.

However, the definitive support for this model will require the affinity measurements between the two proteins.

## SEL1L-HRD1 interaction is required in forming a functional HRD1 ERAD complex

To gain further insights into the role and importance of SEL1L in HRD1 ERAD, we performed unbiased proteomics LC-MS analyses to map SEL1L and HRD1 interactomes in HEK293T cells. Following validation of HRD1 and SEL1L antibodies for IP (Fig. 8a, b), we performed three independent repeats of SEL1L- and HRD1-IP-MS, applied stringent criteria, using corresponding KO cells as negative controls (Supplementary Fig. 8a, b). A total of 24 and 47 high-confidence hits were identified for SEL1L and HRD1 interacting proteins, respectively (Fig. 8c, d, Supplementary Tables 1 and 2). In line with previous studies[58,59], most of the known components of the SEL1L-HRD1 ERAD complex, including HRD1, SEL1L, OS9, ERLEC1, UBE2J1, DERL2, HERP1, FAM8A1 and VCP (Supplementary Fig. 8c), were identified (Supplementary Tables 1 and 2), validating our experimental system. We next further interrogated the HRD1-interactome (24 hits) into two groups based on whether the interaction was SEL1L-dependent (Group I) or not (Group II) (Fig. 8e, f).

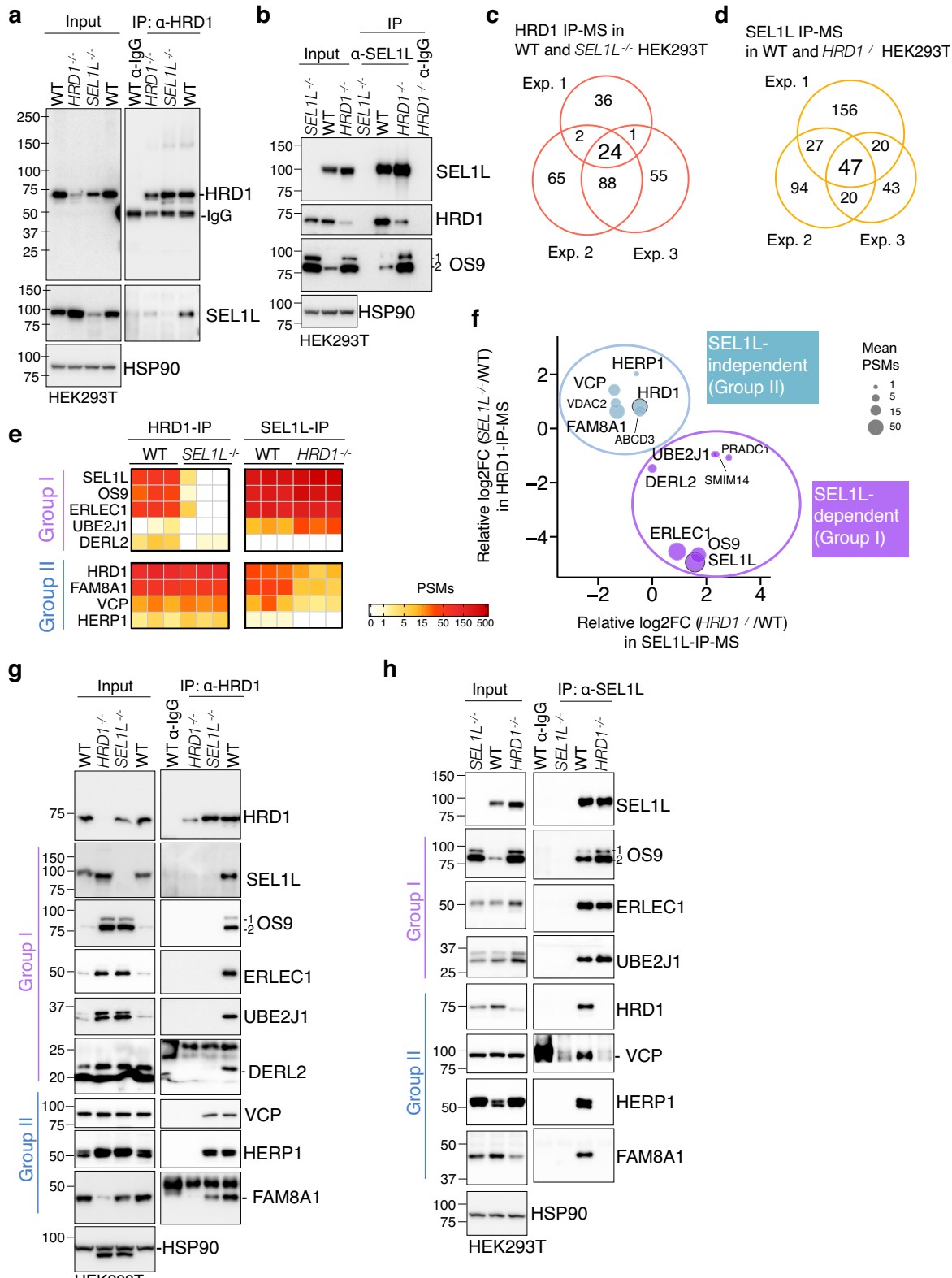

**Fig. 8 | Proteomics identification of SEL1L and HRD1 interactomes. a, b** Immunoprecipitation of HRD1 (**a**) and SEL1L (**b**) in WT, *HRD1*⁻/⁻ and *SEL1L*⁻/⁻ HEK293T cells to validate the HRD1 antibody or home-made SEL1L antibody for IP (*n* = 3 independent repeats per group). **c, d** Venn diagram showing HRD1- and SEL1L- interacting proteins identified in three independent experiments of HRD1- (**c**) and SEL1L- (**d**) IP-MS. **e** Heatmaps of known SEL1L-HRD1 ERAD components from the IP-MS experiments. The proteins were grouped into I and II based on their dependency on SEL1L. **f** Scatter plot showing average log2 fold change (FC) of the PSMs in *HRD1*⁻/⁻ cells compared to WT in SEL1L-IP-MS and of the PSMs in *SEL1L*⁻/⁻ cells compared to WT in HRD1-IP-MS for the 16 overlapping proteins, plus DERL2. Size of the dots is proportional to the mean PSMs in WT samples. **g, h** Immunoprecipitation of endogenous HRD1 (**g**) and SEL1L (**h**) in WT, *HRD1*⁻/⁻ and *SEL1L*⁻/⁻ HEK293T cells to examine their interactions with ERAD components (two independent repeats).

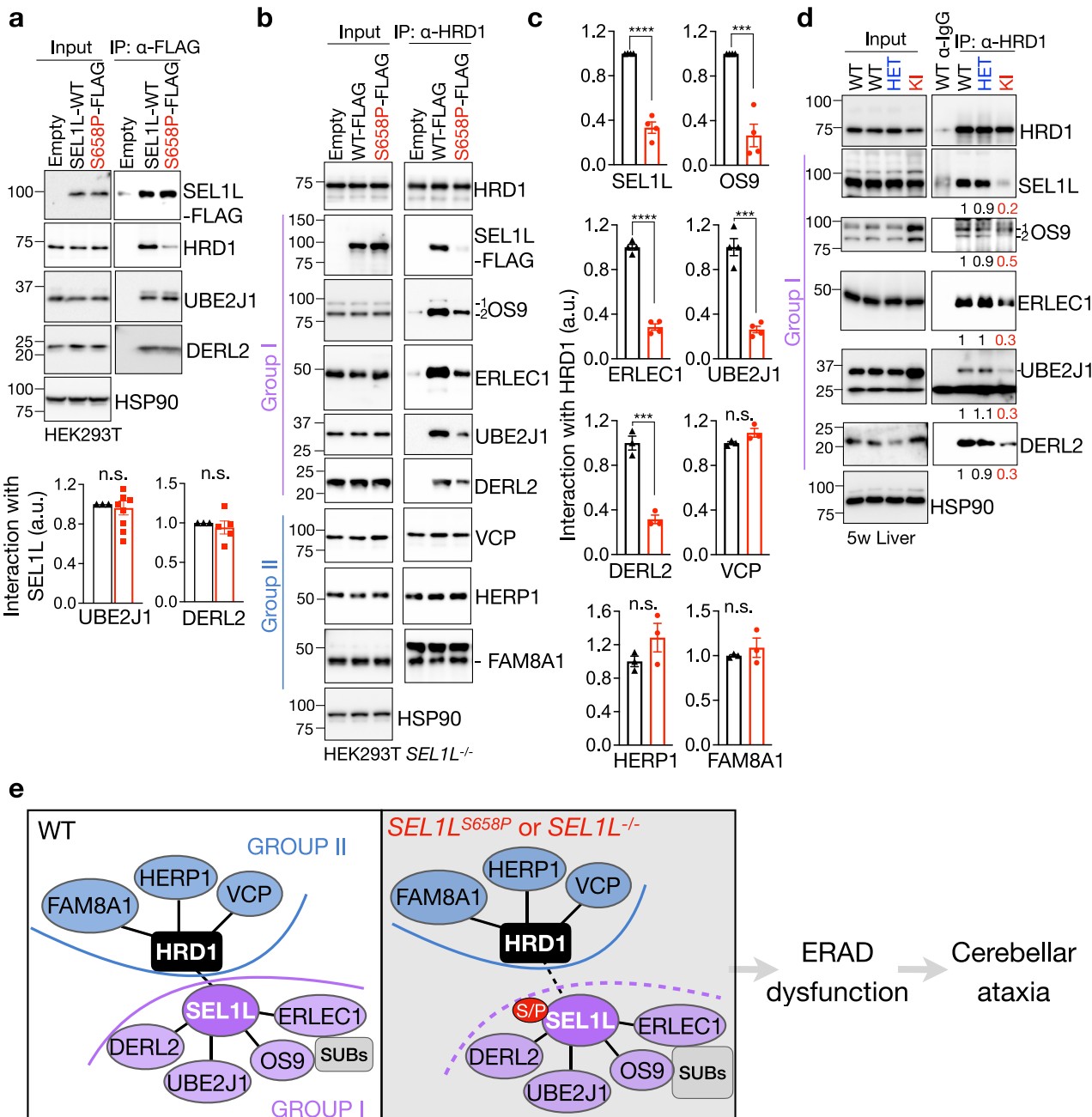

**Fig. 9 | The interaction between SEL1L and HRD1 is a prerequisite for a functional HRD1 ERAD complex. a** Immunoprecipitation of FLAG-agarose in *SEL1L⁻/⁻* HEK293T cells transfected with indicated SEL1L-FLAG constructs to examine the interaction with HRD1, UBE2J1 and DERL2, with quantitation shown below (UBE2J1: $n = 9$ per group; DERL2: $n = 5$ per group; "n" indicates independent samples). **b**, **c** Immunoprecipitation of HRD1 in *SEL1L⁻/⁻* HEK293T cells transfected with indicated SEL1L-FLAG constructs to examine the interaction between endogenous HRD1 and other ERAD components, with quantitation shown in (**c**) (SEL1L/OS9/ ERLEC1/UBE2J1: $n = 4$ per group; DERL2/VCP/HERP1/FAM8A1: $n = 3$ per group; "n" indicates independent samples). **d** Immunoprecipitation of HRD1 in the livers of 5-week-old WT and KI mice to examine the endogenous HRD1 interaction with other ERAD components, with quantitation shown below the gel as means from two independent repeats. HERP1 and FAM8A1 antibodies failed to work in mouse liver tissues. **e** A model showing the role of SEL1L in the formation of HRD1 ERAD complex. S/P indicates S658P. Values, mean ± SEM. n.s. not significant; ***$p < 0.001$ and ****$p < 0.0001$ by two-tailed Student's *t* test (**a**, **c**).

Lectins OS9 and ERLEC1 expectedly formed a tight cluster with SEL1L, i.e. in Group I, through which they interacted with HRD1 (Fig. 8e and purple dots, Fig. 8f). Unexpectedly, UBE2J1 and DERL2 also appeared in Group I (Fig. 8e and purple dots, Fig. 8f), suggesting that their interaction with HRD1 may also be mediated through SEL1L. Mammals have three DERLIN proteins, with DERL2 primarily associated with the SEL1L-HRD1 complex[17,59,70]. Consistent with this, DERL2 was the only DERLIN protein identified in our IP-MS. On the other hand, three known ERAD components (HERP1, VCP and FAM8A1) and two

new hits (VDAC2 and ABCD3) appeared in Group II, where their interactions with HRD1 were SEL1L-independent (Fig. 8e and blue dots, Fig. 8f). IP of endogenous SEL1L and HRD1 followed by Western blot analysis further validated SEL1L dependency for these two groups of HRD1 interactors (Fig. 8g, h).

We next experimentally tested whether the ERAD complex formation is impaired in cells expressing *SEL1L^{S658P}*. Indeed, *SEL1L^{S658P}* attenuated its interaction with HRD1, but not with OS9, ERLEC1 (Fig. 6e), UBE2J1 and DERL2 (Fig. 9a). By contrast, HRD1 interactions

with these Group I proteins were attenuated by 3-4 folds in *SEL1L*<sup>S658P</sup>-expressing cells (Fig. 9b, c). On the other hand, HRD1 interactions with the Group II proteins (e.g. HERP1, VCP and FAM8A1) were unaffected by *SEL1L*<sup>S658P</sup> expression (Fig. 9b, c). In line with the overexpression experiments, HRD1 interactions with Group I proteins were reduced in *SEL1L*<sup>S658P</sup> KI livers, compared to those in WT and HET livers (Fig. 9d). Hence, we conclude that SEL1L is indispensable for HRD1 ERAD function by recruiting Group I factors to HRD1 to form a functional HRD1 ERAD complex (Fig. 9e).

## Discussion

SEL1L-HRD1 ERAD plays a critical role in clearing misfolded proteins in the ER. While ample studies have shown that, individually, they are indispensable in vivo, one key question is whether they always work together or have their own function other than ERAD, or in the case of HRD1, whether it can function without SEL1L as the SEL1L homolog in yeast Hrd3p is dispensable for the interaction between Hrd1 and substrates or E2 enzyme[12]. Here we provide definitive evidence, powered by in vivo mouse models, rigorous mutagenesis and proteomic analyses, that SEL1L interaction with HRD1 is pathologically important and biochemically significant. SEL1L is not only important for HRD1 protein stability[6,10,11], but also all aspects of HRD1 ERAD function, including substrate recruitment, ubiquitination and retrotranslocation (Fig. 9e). These unexpected findings not only demonstrate the pathological significance of this ERAD complex in a whole organism (rather than cell type-specific mouse models), but also offer novel insights into the assembly of a functional ERAD complex.

The importance of SEL1L-HRD1 interaction in vivo remained unknown. Here we show that impaired SEL1L-HRD1 interaction is sufficient to drive disease pathogenesis including partial embryonic lethality, developmental delay, microcephaly and early-onset ataxia in mice as shown in the affected Finnish Hounds as previously reported[62]. *SEL1L*<sup>S658P</sup> KI mice suffering from early-onset ataxia are associated with a mild reduction in Purkinje cell numbers. Purkinje cells are one of the largest neurons with a prominent ER structure that form an interconnected network regulating calcium signaling, protein synthesis, folding, and protein trafficking[71]. Previous studies have shown that Purkinje cells are vulnerable for alterations in ER homeostasis[72,73]. Here we show that SEL1L expression in the cerebellum is highly enriched in Purkinje cells and that expression of *SEL1L*<sup>S658P</sup> variant is associated with mild ER stress in the cerebellum, without causing cell death. Additionally, there was no obvious changes in other neurons, astrocytes and microglia in the cerebellum of KI mice. These data suggest that the impact of this variant on Purkinje cells and ataxia development is likely mediated through substrate(s)-related mechanisms, rather than through ER stress or cell death as previously proposed[62]. The molecular nature of the potential ERAD substrates in Purkinje cells requires further investigation.

Providing further support for the subtle UPR associated with the variant, examination of peripheral tissues of KI mice including the pancreas, livers, kidneys and adipose tissues revealed no obvious histological or functional defects. The lack of an overt UPR in the KI mice is likely due to various adaptive mechanisms in response to a hypomorphic variant, including, but not limited to, the upregulation of ER chaperones to increase folding efficiency, enhanced aggregation and sequestration of misfolded proteins to hence attenuate proteotoxicity, the activation of ER-phagy to clear protein aggregates, and/or ERAD involving alternative E3 ligase to clear misfolded proteins in the ER[53,74,75].

Previous studies of mammalian SEL1L-HRD1 ERAD complex using overexpression systems have showed that overexpressing SEL1L or HRD1 can pull down OS9, ERLEC1, UBE2J1, HERP1, FAM8A1 and DERL2[17,57–59]. An earlier study showed that SEL1L knockdown did not impact the HRD1-UBE2J1 interaction, while loss of HRD1

disrupted the SEL1L-UBE2J1 interaction[57], suggesting that HRD1 directly interacts with UBE2J1. By contrast, our study identified proteome-wide SEL1L-dependent HRD1 interactors which included UBE2J1 and DERL proteins. Indeed, HRD1 interaction with UBE2J1 and DERL was attenuated in cells expressing *SEL1L*<sup>S658P</sup> variant or entirely abolished, hence expanding the role and importance of SEL1L in HRD1 ERAD.

Using a combination of AI structural prediction and biochemical analyses, we showed that SEL1L<sup>S658P</sup> attenuates the SEL1L-HRD1 interaction via an aromatic-aromatic interaction of SEL1L F668 and HRD1 Y30. While the structure of the SEL1L-HRD1 complex is similar to the yeast Hrd3p-Hrd1p complex, none of these residues is present in yeast Hrd3p. This finding suggests that, contrary to the yeast Hrd3p-Hrd1p interaction, mammalian SEL1L may play an additional role in the SEL1L-HRD1 interaction. This study may provide new framework for future therapeutic designs aiming at targeting the SEL1L-HRD1 interaction and function using small molecules.

## Methods

### Institutional approvals

All animal procedures were approved by the Institutional Animal Care and Use Committee of the University of Michigan Medical School (PRO00010658) and University of Virginia (4459-08-23) in accordance with the National Institutes of Health (NIH) guidelines.

### Mice

*SEL1L*<sup>S658P</sup> KI mice (human SEL1L p.Ser658 is equivalent to mouse SEL1L p.Ser654) were generated at the University of Michigan Molecular Genetics Core using the CRISPR-Cas9 technology. The two single guide RNAs (sgRNAs) were designed by using computer algorithm (http://crispor.tefor.net), targeting mouse genome *SEL1L* exon 19 where the mutation is located: sgRNA1: 5′-CAGGCGGTAATGAATAAATGCGG-3′; sgRNA2: 5′-CTGGGCTACATGCACGAGAAGGG -3′. The sgRNAs were synthesized using the Synthego sgRNA synthesis Kit per manufacturer's protocol, tested and confirmed in fertilized eggs. A donor DNA carrying mouse *Sel1L* cDNA 1960C>T mutation and additional silent mutations was designed to mediate homology-directed repair (HDR): 5′-GTTACACTGTGGCTAGAATTAAGCTTGGAGACTACCACTT CTATGGCTTTGGCACTGATGTGGATTATGAGACAGCCTTCATTCACTA CCGGCTGGCTCCTGAGCAGCAGCACAGCGCCCAAGCTATGTTTAACC TGGGATACATGCATGAGAAAGGCCTAGGCATTAAACAGGTGAGTGTG GGCCACGCTGGCGCTGAG-3′. The donor DNA was synthesized as Ultramer dsDNA by Integrated DNA Technologies, Inc. A mixture of Cas9 protein (Sigma), sgRNAs, and donor DNA was microinjected into fertilized mouse eggs on the B6/SJL background. The injected zygotes were then transferred into pseudopregnant females. A pair of primers was designed for genotyping: Ser658Pro-F (5′-ATTCACTACCGGCTG GCTC-3′) and WT-R (5′-TCCTGTAGCCCGAGGTCTGA-3′). Tail samples were obtained from 2-week-old pups for genotyping and heterozygous *SEL1L*<sup>S658P</sup> KI mice were established as F0 founders. The existence of desired mutations was further confirmed using the Sanger sequencing in two independent founders. The founders were then bred separately to WT C57BL6/J mice to obtain F1 heterozygous *SEL1L*<sup>S658P</sup> KI mice. F1 heterozygous *SEL1L*<sup>S658P</sup> mice were inter-crossed to generate homozygous *SEL1L*<sup>S658P</sup> KI mice, WT and heterozygous littermates.

To study embryonic lethality, pregnant heterozygous female mice were euthanized at embryonic day 10-12 (E10-12) or at E14-16. Uterine horns were removed and placed them in a petri dish filled with cold PBS. The embryos were removed from the amniotic membrane and transferred into a new petri filled with cold PBS. A small tail biopsy was used for genotyping. The post-natal lethality data were obtained from 71 litters of 28 breeding pairs total. Age- and sex-matched littermates were maintained in a temperature-controlled room on a 12-h light/dark cycle and used in all studies.

## Genotyping

Mice were routinely genotyped using PCR of genomic DNA samples obtained from tails or ears with the following primer pairs:

$SEL1L^{S658P}$ allele for $SEL1L^{S658P}$ KI mice: F: 5'-ATTCACTACCGGCTGGCTC-3'; R: 5'-TCCTGTAGCCCGAGGTCTGA-3';

$SEL1L$ wildtype allele for $SEL1L^{S658P}$ KI mice: F: 5'-CGCATTTATTCATTACCGCCTG-3'; R: 5'-TCCTGTAGCCCGAGG TCTGA-3';

## Behavioral studies

All behavior procedures were performed by investigators blind to the genotype of each group. The indirect calorimetry (including food intake) was performed and analyzed at University of Michigan Animal Phenotyping Core as previously described[76,77]. Food intake was measured in single-housed mice on a regular chow diet at room temperature (20–23 °C). The measurements were carried out continuously for 72 h.

For the hindlimb clasping assessment, mice were lifted up by tails and held over a cage for 1 min to assess abnormal hind limb clasping and scored as previously described[78]. The mice were then given a score ranging from 0 to 4 based on the following criteria:

0 - No limb clasping

1 - One hindlimb splayed incompletely; toes splayed normally

2 - Both hindlimbs splayed incompletely; toes splayed normally

3 - Both hindlimbs clasped and both forelimbs clasped; toes curled

4 - Forelimbs and hindlimbs all clasped together; toes curled

For the gait analysis, hind- and fore-paws were coated with red and black nontoxic food coloring, respectively. Mice were then allowed to walk along a 450-mm-long, 50-mm-wide runway with 70-mm-high walls with white paper lining the floor, into a darkened enclosed escape box. The footprints were analyzed for stride length (the distance covered by the same hind paw), the stride width (the distance from one hind-limb that intersects perpendicularly with the line for stride length on the contralateral hind paw), paw matched distance (the distance between hind and forepaw) and the ratio between stride length and width. All mice received three trainings and a trial run.

The balance beam study was used to evaluate motor coordination and balance as previously described[79]. Mice were trained to stay upright and walk across an elevated narrow beam to a safe platform for two consecutive days (three times per day). The beam apparatus consisted of a flat beam that was 1-meter long and 12 mm wide, resting 50 cm above the table on top of two poles. A darkened escape box with nesting material was placed at the end of the beam as the finish point. On the third day, the time to cross a distance of 80 cm located in the center of the beam was measured. A video camera was set on a tripod to record the performance of the animals during the test. Room temperature, humidity, lighting, and background noise were kept consistent throughout the experiment.

For the rotarod test, mice were placed into a rod rotating (Model LE8500, Panlab SL) at an accelerating speed. Animals were tested in four trials per day for three consecutive days. The tests started with 4 rpm and increased to 40 rpm after 300 s; the time and the maximal speed until mice fell from the rod were measured.

## Histology and immunofluorescence

Anesthetized mice were perfused with 20 ml of 0.9% NaCl followed by 40 ml of 4% paraformaldehyde in 0.1 M PBS pH 7.4 for fixation. Tissues were dissected out and fixed overnight in 4% paraformaldehyde in PBS at 4 °C. For hematoxylin and eosin (H&E) staining, samples were dehydrated, embedded in paraffin, and stained at the Rogel Cancer Center Tissue and Molecular Pathology Core or In-Vivo Animal Core at the University of Michigan Medical School. Quantitation of Purkinje cell numbers was performed on H&E stained sections. Counts were normalized to the length of the Purkinje cell layer, as measured by Aperio ImageScope software, and reported as Purkinje cell density. For immunofluorescence staining, paraffin-embedded brains sections were deparaffinized in xylene and rehydrated using graded ethanol series (100%, 90%, 70%), followed by rinse in distilled water. Antigen retrieval was performed by boiling the slides in a microwave in sodium citrate buffer. Sections were then incubated in a blocking solution (5% donkey serum, 0.3% Triton X-100 in PBS) for 1 h at room temperature and with primary antibodies (Calbindin, Cell Signaling, #2173, 1:100; KDEL, Novus Biologicals, #97469, 1:200; NeuN, Sigma, #ABN90, 1:200; GFAP, Cell Signaling, #3670, 1:100; IBA1, Fujifilm, #019-19741, 1:100; SEL1L, home-made, 1:300) overnight at 4 °C in a humidifying chamber. The next day, following 3 washes with PBST (0.03% Triton X-100 in PBS), slides were incubated with the respective Alexa Fluor–conjugated to secondary antibodies (Jackson ImmunoResearch; dilution 1:500) for 1 h at room temperature, followed by mounting with VECTASHIELD mounting medium containing DAPI (Vector Laboratories, H-1500). Images were captured using the Nikon A1 confocal microscope at the University of Michigan Morphology and Image Analysis Core.

## Serum chemistry tests

Mouse blood was collected without anticoagulation via cardiac puncture immediately prior to sacrifice in mice anesthetized by isoflurane inhalation. The whole blood was allowed to clot under room temperature for 15–30 min and then serum was separated by centrifugation. Serum Mini Chemistry Panel Tests (Alanine Transaminase, Total Bilirubin, Alkaline Phosphatase, Blood Urea Nitrogen, Creatinine) were performed at the Unit for Laboratory Animal Medicine (ULAM) In Vivo Animal Core pathology laboratory at the University of Michigan.

## 3D MRI brain analysis

Anesthetized mice were transcardially perfused with 4% paraformaldehyde in PBS. The head were cut and removed the skin and soft tissue and then soaked in PBS containing 5% Gd-DTPA and stored in 4 °C for 5 days. Prior to imaging, the brain specimens were placed in a 15 ml tube containing proton signal-free susceptibility-matched fluid (Galden Heat Transfer Fluid, HT230. SOLVEY, Italy), which was placed in a mouse head coil. MRI acquisitions were performed on a 7T simultaneous PET-MR scanner (MR Solutions Ltd.) at the Zilkha Neurogenetic Institute (University of Southern California). The 2D sagittal T2-weighted images (T2WI) were performed using fast spin echo (FSE) sequence (TR/TE: 4000/45 ms, echo train length of 7, eight average, 28 slices, slice thickness of 500 μm, in-plane resolution $390 \times 140$ μm³), to manually draw the regions of interests (ROIs) of cerebellum and cortex. 3D T2Wis were performed using Fast Low Angle Shot (FLASH) sequence (TR/TE: 50/5 ms, FA = 30°, seven average, with an isotropic voxel resolution of $93 \times 93 \times 93$ μm³) for 3D brain render reconstruction For the reconstruction, 3D images were performed using our in-house MATLAB code. The intensity inhomogeneity of images was corrected by N4 algorithm and then co-registered to the Mouse Magnetic Resonance Microscopy Atlas (https://www.loni.usc.edu/research/atlas_downloads). The masks of whole brain, cerebellum and cortex were segmented and manually edited using the ITK-SNAP software. The rendered 3D brain was reconstructed, and masks of cerebellum and cortex were displayed using the ParaView software version 5.10.1.

## CRISPR/Cas9-based knockout (KO) and KI HEK293T cells

HEK293T cells were originally obtained from ATCC and cultured at 37 °C with 5% $CO_2$ in DMEM with 10% fetal bovine serum (Fisher Scientific). To generate SEL1L- and HRD1- deficient HEK293T cells, sgRNA oligonucleotides designed for human $SEL1L$ (5'-GGCTGAACAGGGCTATGAAG-3') or human $HRD1$ (5'-GGACAAAGGCCTGGATGTAC-3') was inserted into lentiCRISPR v2 (plasmid 52961; Addgene). Cells grown in 10 cm petri dishes were transfected with indicated plasmids using 5 μl 1 mg/ml polyethylenimine (PEI, Sigma) per 1 μg plasmids for

HEK293T cells. Cells were cultured 24 h after transfection in medium containing 2 µg/ml puromycin for 24 h followed by normal growth media.

*SEL1L*[S658P] KI HEK293T cells was generated using the CRISPR-Cas9 system and homology-directed repair (HDR) mechanism (Integrated DNA Technologies, IDT). 5 µl 100 µM Alt-R crRNA (IDT) with gRNA sequence was first mixed with 5 µl 100 µM Alt-R tracrRNA (IDT) containing Cas9 interacting sequence. To anneal the oligos, the duplex mixture was heated at 95 °C for 5 min and then cooled at room temperature for 20 min. 9 µl of the guide complex was incubated with 6 µl 62 µM Alt-R Cas9 enzyme (IDT) at room temperature for 20 min. 5 µl of the ribonucleoprotein (RNP) complex, together with 1.2 µl 100 µM HDR Donor Oligo (IDT), 1.2 µl 100 µM Alt-R Cas9 Electroporation Enhancer (IDT), was mixed with 100 µl HEK293T cell suspension (about 5 × 10⁵ cells) in Electroporation Solution (Ingenio). The mixture was transferred into a 0.2 cm cuvette and electroporation was performed using Lonza Nucleofector IIb (Lonza). To prepare the cell culture media, 3.4 µl 0.69 mM Alt-R HDR Enhancer V2 (IDT) was added to 2000 µl DMEM with 10% fetal bovine serum (Fisher Scientific). After electroporation, the cell suspension was added to the cell culture media, and the mixture was incubated in 4 wells of a 24-well dish (500 µl per well). The cells were cultured at 37 °C with 5% CO2. After 5 days of incubation, the genomic DNA of the cell culture was extracted with 50 mM NaOH. DNA fragments covering the target sites were amplified by PCR, using HotStart Taq 2X PCR Master (ABclonal), and analyzed using the Sanger Sequencing (Eurofins) to estimate the percentage of mutant allele in the cell pool. In parallel, cell culture was diluted into 8 cells per mL and cultured in eight 96-well plates (100 µl per well) for single cell isolation. After 10 days, 100 single cell colonies were transferred into 24-well plates. The *SEL1L*[S658P] region was amplified using a 25 µl PCR reaction and sequenced. Cell colonies with homozygous *SEL1L*[S658P] alleles were transferred into a 6-well plate for further experiments.

crRNA (guide sequence): 5'- GTTGCTGCTCAGAAGCCAGA-3'
HDR Donor Oligo (the mutation site is underlined): 5'-CCCA-GATTAAACATAGCTTGTGCACTGTGTTGCTGCTCAG<u>G</u>AGCCAGA CGGTAATGAATAAATGCAGTTTCATAATCTACATCG<u>G</u>-3'
Amplification PCR primers:
F: 5'-TGTAAGCATGTCAATGGGAGGAG-3'
R: 5'- AGCGTGGATAGCAGTATCTGT-3'
Sequencing primers:
5'- GTTTAAGGCTATACTGTGGCT-3'

## Protein structure modeling

The individual structure models of human SEL1L, HRD1, OS9, and DERLIN1 were downloaded from AlphaFold2 database (https://alphafold.ebi.ac.uk/)[80]. The complex structure was constructed using TM-align[81] by superposing the predicted human SEL1L, HRD1, OS9, and DERLIN1 structures to the CryoEM structure of the yeast Hrd3p-Hrd1p-Der1p protein complex (PDB ID: 6VJZ) and Hrd3p-Yos9 complex (PDB ID: 6VK3). All the structure images of human SEL1L 171-723 aa, HRD1 1-334 aa, OS9 33-655 aa and DERLIN1 1-213 aa were rendered using PyMOL (version 2.3.2). To analyze the evolutionary conservation of each residue, a position-specific scoring matrix (PSSM) was generated from a PSI-BLAST search of the target protein through the NCBI NR database[82,83].

The amino acid sequence of human SEL1L (accession no. NP_005056.3) was aligned with SEL1L homologs from chimpanzee (JAA44458.1), gibbon (XP_003260889.1), marmoset (XP_002754215.1), dog (XP_038530327.1), rabbit (XP_002719662.2), hamster (NP_001268291.1), mouse (NP_001034178.1), bird (XP_021143193.1), frog (XP_041430335.1), bony fish (NP_001038629.1), shark (XP_048393621.1), drosophila (NP_001262882.1) and yeast (QHB10358.1) using the ClustalW program.

The amino acid sequence of human HRD1 (SYVN1) (accession no. NP_115807.1) was aligned with HRD1 homologs from chimpanzee (JAA39943.1), gibbon (XP_032009276.1), marmoset (JAB46248.1), dog (XP_038280973.1), hamster (XP_051049706.1), mouse (AAH46829.1), bird (XP_053824008.1), frog (XP_012816159.1), bony fish (AAH44465.1), shark (XP_043538543.1), drosophila (NP_001263152.1) and yeast (QHB11597.1) using the ClustalW program.

## Plasmids

The following plasmids were used in the study: (h denotes human genes; m denotes mouse genes): *mSel1L* cDNA was cloned from mouse liver cDNA and inserted into the pcDNA3 to generate pcDNA3-mSEL1L(WT)-FLAG; pcDNA3-h-HRD1-Myc and pcDNA3-h-proAVP(G57S)-HA were described previously[36]; The SEL1L-FLAG mutants S658P, S658A, S658D, S658K, S663P, F668A, F668S, F668T, F668Y, F668M, F668D, F668K, F668H, F668N, F668W and F668C were generated using the plasmid pcDNA3-mSEL1L(WT)-FLAG as the template. P658S and the double mutations of SEL1L (S658P mutated back to WT) was generated using the plasmid pcDNA3-mSEL1L(S658P)-FLAG. All plasmids were validated by DNA sequencing.

The primers are:
mSEL1L-FLAG-F: 5'-CGCGGATCCACCATGCAGGTCCGCGTCAGGC TGTCG-3'
R: 5'-CGCTCTAGACTATTTATCATCATCATCTTTATAATCTCCGCC CTGTG
GTGGCTGCTGCTCTGG-3'
S658P-F: 5'-GCATTTATTCATTACCGCCTGGCTCCTGAGCAGC-3'
R: 5'-GCTGCTCAGGAGCCAGGCGGTAATGAATAAATGC
S658A-F: 5'-GCATTTATTCATTACCGCCTGGCTGCTGAGCAGC-3'
R: 5'-GCTGCTCAGCAGCCAGGCGGTAATGAATAAATGC-3'
S658D-F: 5'-GCATTTATTCATTACCGCCTGGCTGATGAGCAGC-3'
R: 5'-GCTGCTCATCAGCCAGGCGGTAATGAATAAATGC-3'
S658K-F: 5'-GCATTTATTCATTACCGCCTGGCTAAAGAGCAGC-3'
R: 5'-GCTGCTCTTTAGCCAGGCGGTAATGAATAAATGC-3'
P658S-F: 5'-GCATTTATTCATTACCGCCTGGCTTCTGAGCAGC-3'
R: 5'-GCTGCTCAGAAGCCAGGCGGTAATGAATAAATGC-3'
S663P-F: 5'-CTTCTGAGCAGCAGCACCCCGCCCAAGCTATG-3'
R: 5'-CATAGCTTGGGCGGGGTGCTGCTGCTCAGAAG-3'
F668A-F: 5'-GCCCAAGCTATGGCTAACCTGGGCTAC-3'
R: 5'-GTAGCCCAGGTTAGCCATAGCTTGGGC-3'
F668S-F: 5'-GCCCAAGCTATGTCTAACCTGGGCTAC-3'
R: 5'-GTAGCCCAGGTTAGACATAGCTTGGGC-3'
F668T-F: 5'-GCCCAAGCTATGACCAACCTGGGCTAC-3'
R: 5'-GTAGCCCAGGTTGGTCATAGCTTGGGC-3'
F668Y-F: 5'-GCCCAAGCTATGTACAACCTGGGCTAC-3'
R: 5'-GTAGCCCAGGTTGTACATAGCTTGGGC-3'
F668M-F: 5'-GCCCAAGCTATGATGAACCTGGGCTAC-3'
R: 5'-GTAGCCCAGGTTCATCATAGCTTGGGC-3'
F668D-F: 5'-GCCCAAGCTATGGATAACCTGGGCTAC-3'
R: 5'-GTAGCCCAGGTTATCCATAGCTTGGGC-3'
F668K-F: 5'-GCCCAAGCTATGAAGAACCTGGGCTAC-3'
R: 5'-GTAGCCCAGGTTCTTCATAGCTTGGGC-3'
F668H-F: 5'-GCCCAAGCTATGCACAACCTGGGCTAC-3'
R: 5'-GTAGCCCAGGTTGTGCATAGCTTGGGC-3'
F668N-F: 5'-GCCCAAGCTATGAATAACCTGGGCTAC-3'
R: 5'-GTAGCCCAGGTTATTCATAGCTTGGGC-3'
F668W-F: 5'-GCCCAAGCTATGTGGAACCTGGGCTAC-3'
R: 5'-GTAGCCCAGGTTCCACATAGCTTGGGC-3'
F668C-F: 5'-GCCCAAGCTATGTGTAACCTGGGCTAC-3'
R: 5'-GTAGCCCAGGTTACACATAGCTTGGGC-3'
hHRD1 cDNA was cloned from pcDNA3-hHRD1(WT)-Myc-His (a kind gift from Y. Ye, NIDDK, USA) and inserted into the pcDNA3 to generate pcDNA3-hHRD1(WT)-FLAG. Point mutations of HRD1 in this study were also generated using site-directed mutagenesis. The HRD1-

FLAG mutants Y30A, Y30D, Y30K and Y30F were generated using the plasmid pcDNA3-hHRD1(WT)-FLAG as the template.

hHRD1-FLAG-F: 5′- GGC GGTACC ATGTTCCGCACGGCAGTGAT-GATG −3′, R: 5′- GGC GGATCC TCATTTATCATCATCATCTTTA-TAATCTCCGCCGTGGGCAACAGGAGACTC −3′.

Y30A-F: 5′-CACCAGTTCGCCCCCACTGTGGTG-3′
R: 5′-CACCACAGTGGGGGCGAACTGGTG
Y30D-F: 5′-CACCAGTTCGACCCCACTGTGGTG
R: 5′-CACCACAGTGGGGTCGAACTGGTG
Y30K-F: 5′-CACCAGTTCAAGCCCACTGTGGTG
R: 5′-CACCACAGTGGGCTTGAACTGGTG
Y30F-F: 5′-CACCAGTTCTTCCCCACTGTGGTG
R: 5′-CACCACAGTGGGGAAGAACTGGTG

## Western blot and antibodies

Mouse tissues or HEK293T cells were harvested and snap-frozen in liquid nitrogen. The proteins were extracted by sonication in NP-40 lysis buffer (50 mM Tris-HCl at pH7.5, 150 mM NaCl, 1% NP-40, 1 mM EDTA) with protease inhibitor (Sigma), DTT (Sigma, 1 mM) and phosphatase inhibitor cocktail (Sigma). Lysates were incubated on ice for 30 min and centrifuged at 16,000 g for 10 min. Supernatants were collected and analyzed for protein concentration using the Bio-Rad Protein Assay Dye (Bio-Rad). 20–50 μg of protein were denatured at 95 °C for 5 min in 5x SDS sample buffer (250 mM Tris-HCl pH 6.8, 10% sodium dodecyl sulfate, 0.05% Bromophenol blue, 50% glycerol, and 1.44 M β-mercaptoethanol). Protein was separated on SDS-PAGE, followed by electrophoretic transfer to PVDF (Fisher Scientific) membrane. The blots were incubated in 2% BSA/Tri-buffered saline tween-20 (TBST) with primary antibodies overnight at 4 °C: anti-HSP90 (Santa Cruz, #sc-13119, 1:5000), anti-SEL1L (home-made, 1:10,000)[44], anti-HRD1 (Proteintech, #13473-1, 1:2000), anti-OS9 (Abcam, #ab109510, 1:5000), anti-CD147 (Proteintech, #11989-1, 1:3000), anti-IRE1α (Cell Signaling, #3294, 1:2000), anti-ERLEC1 (Abcam, #ab181166, 1:5000), anti-UBE2J1 (Santa Cruz, #sc-377002, 1:3000), anti-DERL2 (gift from Chih-Chi Andrew Hu, 1:1000), anti-BiP/GRP94 (Abcam, #ab21685, 1:5000), anti-PDI (Enzo, #ADI-SPA-890, 1:5000), anti-FLAG (Sigma, #F1804, 1:1000), anti-HA (Sigma, #H3663, 1:5000), anti-Myc (Sigma, #C3956, 1:3000), anti-Pro-Caspase-3 (Cell Signaling, #9662, 1:2000), anti-cleaved-Caspase-3 (Cell Signaling, #9661, 1:1000), anti-Calbindin (Cell Signaling, #2173, 1:5000), anti-PERK (Cell Signaling, #3192, 1:5000), anti-p-PERK (Cell Signaling, #3179, 1:1000), anti-eIF2α (Cell Signaling, #9722, 1:5000), anti-p-eIF2α (Cell Signaling, #9721, 1:1000), anti-VCP (Proteintech, #10736-1-AP, 1:3000), anti-HERP1 (Abcam, #ab150424, 1:3000), anti-FAM8A1 (Proteintech, #24746-1-AP,1:3000), anti-CHOP (Cell Signaling, #2895S, 1:1000), anti-GFAP (Cell Signaling, #3670S, 1:3000), anti-IbaI (Proteintech, #10904-1, 1:3000). Membranes were washed with TBST and incubated with secondary antibodies, either HRP conjugated (Bio-Rad, 1:10,000), anti-Rabbit IgG True-Blot HRP (Rockland, #18-8816-33, 1:500) or anti-Mouse IgG TrueBlot-HRP (Rockland, #18-8817-31, 1:500) at room temperature for 1 h prior to the ECL chemiluminescence detection system (Bio-Rad) development. Band intensity was determined using Image lab (Bio-Rad) software.

## Phos-tag gels

IRE1α phosphorylation was measured by a Phos-tag-based Western blot method[84]. Phos-tag gel was carried out the same way as regular Western blots, except that: (a) 5% SDS-PAGE containing 50 μM Phos-tag (NARD Institute) and 50 μM Phos-tag MnCl₂ (Sigma) were used; (b) gels were soaked in 1 mM EDTA for 10 min prior to transfer onto a PVDF membrane.

## Generation of SEL1L- and HRD1- specific antibodies

The home-made SEL1L and HRD1 antibodies were generated by Abclonal Technology. Rabbit polyclonal anti-human SEL1L and HRD1 antibodies were raised in rabbits immunized with glutathione S-transferase (GST) fused N-terminal 172 amino acids (23–194 aa) of human SEL1L and N-terminal 241 amino acids (310–550 aa) of human HRD1, respectively. The polyclonal antibody was generated by immunizing rabbits with the recombinant hSEL1L or hHRD1 proteins, and further affinity-purified using the same antigen.

## Cycloheximide (CHX) treatment of HEK293T cells

HEK293T cells were cultured in the cell culture medium with 50 μg/ml CHX for the indicated time and snap-frozen in liquid nitrogen for protein extraction.

## Immunoprecipitation (IP)

HEK293T cells or the livers were snap-frozen in liquid nitrogen and whole cell lysate was prepared in IP lysis buffer [For IP in cells: 150 mM NaCl, 0.2% Nonidet P-40 (NP40), 0.1% Triton X-100, 25 mM Tris-HCl pH 7.5; For IP in liver: 150 mM NaCl, 1% Nonidet P-40 (NP40), 25 mM Tris-HCl pH 7.5] supplemented with protease inhibitors, protein phosphatase inhibitors, and 10 mM N-ethylmaleimide. A total of ~5 mg protein lysates were incubated with 10 μl anti-FLAG agarose (Sigma, #A2220), 10 μl Myc agarose (Sigma, #7470), 1 μl anti-SEL1L or anti-HRD1 home-made antibody overnight at 4 °C with gentle rocking. SEL1L or HRD1-IP lysates were incubated with Protein A agarose (Invitrogen, #20333) at 4 °C for 2 h. The incubated agaroses were washed three times with IP lysis buffer and eluted in SDS sample buffer at 95 °C for 5 min followed by SDS-PAGE and Immunoblot.

## (Non-)reducing SDS-PAGE

Whole cell lysates were prepared in NP-40 lysis buffer supplemented with protease and phosphatase inhibitors and 10 mM N-ethylmaleimide. Lysates were incubated on ice for 30 min and centrifuged at 16,000 × g for 10 min. Supernatants were collected and analyzed for protein concentration using the Bio-Rad Protein Assay Dye (Bio-Rad). For reducing SDS-PAGE analysis, lysates were denatured at 95 °C for 5 min in the 5x SDS sample buffer. For non-reducing SDS-PAGE analysis, the lysates were prepared in 5x non-denaturing sample buffer (250 mM Tris-HCl pH 6.8, 1% sodium dodecyl sulfate, 0.05% bromophenol blue, 50% glycerol) and incubated at 37 °C for 1 h. The samples were loaded on a 4%-12% gradient SDS-PAGE.

## RNA preparation and RT-PCR

Total RNA was extracted from tissues using TRI Reagent and BCP phase separation reagent (Molecular Research Center, TR 118). The ratio of Xbp1s to total Xbp1 (Xbp1u + Xbp1s) levels was quantified by Image Lab (Bio-Rad) software. For RT-PCR analysis the following primer sequences were used:

mXbp1 F: ACGAGGTTCCAGAGGTGGAG R: AAGAGGCAACAGTGTCAGAG

mL32 F: GAGCAACAAGAAAACCAAGCA R: TGCACACAAGCCATCTACTCA

## Transmission electron microscopy (TEM)

Mice were anesthetized and perfused with 3% glutaraldehyde, 3% formaldehyde in 0.1 M cacodylate buffer (Electron Microscopy Sciences). Cerebellum was cut into small pieces and fixed overnight at 4 °C in 3% glutaraldehyde, 3% formaldehyde in 0.1 M Sorenson's buffer (Electron Microscopy Sciences). The tissues were then embedded and sectioned at the University of Michigan Histology and Imaging Core. Samples were stained with uranyl acetate/lead citrate and high-resolution images were acquired with a JEOL 1400-plus electron microscope (JEOL).

## Mass spectrometry

The SEl1L-IP in HEK293T cells has been described previously[85]. For SEL1L- and HRD1-IP in HEK293T cells, immunoprecipitation of endogenous SEL1L or HRD1 in WT, *SEL1L*[−/−] or *HRD1*[−/−] HEK293T cells were performed using 10 mg proteins from each sample lysed with the NP-40 lysis buffer supplemented with protease inhibitors, protein phosphatase inhibitors, and 10 mM N-ethylmaleimide. The cell lysates were first incubated anti-SEL1L (home-made) or anti-HRD1 (home-made or Cell Signaling #14773) at 4 °C overnight, followed by incubation with protein A agarose (Invitrogen, #20333) at 4 °C for 2 h. Cell lysates of *SEL1L*[−/−] or *HRD1*[−/−] cells incubated with IgG was included as a negative control for SEL1L- or HRD1-IP, respectively.

The beads were resuspended in 50 μl of 0.1 M ammonium bicarbonate buffer (pH-8). Cysteines were reduced by adding 50 μl of 10 mM DTT and incubating at 45 °C for 30 min. Samples were cooled to room temperature and alkylation of cysteines was achieved by incubating with 65 mM 2-Chloroacetamide, under darkness, for 30 min at room temperature. An overnight digestion with 1 μg sequencing-grade modified trypsin was carried out at 37 °C with constant shaking in a Thermomixer. Digestion was stopped by acidification and peptides were desalted using SepPak C18 cartridges using manufacturer's protocol (Waters). Samples were completely dried using vacufuge. Resulting peptides were dissolved in 0.1% formic acid/2% acetonitrile solution and were resolved on a nano-capillary reverse phase column (Acclaim PepMap C18, 2 micron, 50 cm, ThermoScientific) using a 0.1% formic acid/2% acetonitrile (Buffer A) and 0.1% formic acid/95% acetonitrile (Buffer B) gradient at 300 nl/min over a period of 180 min (2–25% buffer B in 110 min, 25–40% in 20 min, 40–90% in 5 min followed by holding at 90% buffer B for 10 min and re-equilibration with Buffer A for 30 min). Eluent was directly introduced into Q exactive HF mass spectrometer (Thermo Scientific, San Jose CA) using an EasySpray source. MS1 scans were acquired at 60 K resolution (AGC target = $3 \times 10^6$; max IT = 50 ms). Data-dependent collision induced dissociation MS/MS spectra were acquired using Top speed method (3 seconds) following each MS1 scan (NCE ~ 28%; 15 K resolution; AGC target $1 \times 10^5$; max IT 45 ms).

Proteins were identified by searching the MS/MS data against UniProt entries using Proteome Discoverer (v2.4, Thermo Scientific). Search parameters included MS1 mass tolerance of 10 ppm and fragment tolerance of 0.2 Da; two missed cleavages were allowed; carbamidomethylation of cysteine was considered fixed modification and oxidation of methionine, deamidation of asparagine and glutamine were considered as potential modifications. False discovery rate (FDR) was determined using Percolator and proteins/peptides with an FDR of ≤1% were retained for further analysis.

SEL1L- and HRD1-interacting proteins were selected based on the peptide spectrum matches (PSMs) from the label-free IP-MS results. For each protein hit, the PSM value from the IgG sample must be smaller than one-tenth of the WT sample from the same experiment; PSMs ratio of the bait KO sample to WT sample must be smaller than the ratio of the corresponding bait; specifically for the hits passed selection in two but not all three experiments, a minimal PSM value of 2 in at least one WT sample is reinforced. Common contaminating proteins in IP experiments, including keratin, keratin-associated proteins, ribosomal proteins and nuclear proteins were excluded. The proteins in association with both SEL1L and HRD1 were identified from the overlapping protein hits passed the selection criteria in SEL1L- and HRD1-immunoprecipitants.

The topology of the interacting proteins in association with the SEL1L-HRD1 complex was mapped based on their dependency on either SEL1L or HRD1. The hits with reduced PSMs in *SEL1L*[−/−] relative to WT in HRD1 IP-MS, and comparable PSM values in WT and *HRD1*[−/−] cells in SEL1L IP-MS were categorized hits matching this pattern as SEL1L-dependent hits, referred to as Group I. The hits with comparable PSM values in WT and *SEL1L*[−/−] cells in HRD1 IP-MS and low *HRD1*[−/−] to WT PSM ratios in SEL1L IP-MS, were classified as HRD1-dependent hits, or Group II

## Statistical analysis

Statistics tests were performed in GraphPad Prism version 8.0 (GraphPad Software). Unless indicated otherwise, values are presented as mean ± standard error of the mean (SEM). All experiments have been repeated at least two to three times and/or performed with multiple independent biological samples from which representative data are shown. Statistical differences between the groups were compared using the unpaired two-tailed Student's *t* test for two groups or one-way ANOVA or two-way ANOVA for multiple groups. $P < 0.05$ was considered statistically significant.

## Reporting summary

Further information on research design is available in the Nature Portfolio Reporting Summary linked to this article.

## Data availability

Proteomics datasets for the HRD1 and SEL1L IP-MS in HEK293T cells are available via ProteomeXchange with identifiers PXD043674 and PXD041882, respectively. The materials and reagents used are either commercially available or available upon request. All other data are available in the main text or in the supplementary information. Source data are provided with this paper.

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

## Acknowledgements

We thank Dr. Chih-Chi Andrew Hu (Houston Methodist Hospital) for reagents; Drs. Ingrid Bergin and Kendra Andrie at the ULAM In-Vivo Animal Core at University of Michigan Medical School for tissue pathological analyses; the Molecular Genetics Core of the Michigan Diabetes Research Center and the Transgenic Animal Model Core for transgenic animal production; the Proteomics Resource Facility, the Microscopy and Image Analysis Core (NIH S10OD28612-01-A1 and P30DK20572), animal phenotyping cores (NIH P30AR069620, P30DK020572, P30DK089503 and 1U2CDK110678-01), and the ULAM In-Vivo Animal Core at University of Michigan Medical School for assistance; and members of the Qi and Arvan laboratories for technical assistance and insightful discussions. This work was supported by RF1NS122060 (Z.Z.), R01DK128077, R01DK132068 (S.S.), R01DK120047, R01DK120330, R35GM130292 and Michigan Protein Folding Disease Initiative (L.Q.). L.L.L. was and Z.J.L. is supported in part by National Ataxia Foundation Post- and Pre-doctoral Fellowships (NAF 918037 and 1036307).

## Author contributions

L.L.L. designed, performed most experiments and wrote the methods and figure legends; X.W. performed the structural analysis; Y.L. performed the proteomics analysis; B.P., H.H.W., M.T., Z.J.L., H.M., H.W. and L.E.Z. assisted with some in vitro and in vivo experiments; X.L. and Z.Z. performed MRI, involved in experimental design and provided insightful discussion; L.Q. and S.S. directed the study and wrote the manuscript with the help from L.L.L; all authors commented on and approved the manuscript.

## Competing interests

The authors declare no competing interests.
