## [Peer Review File · Nature Communications]

SEL1L-HRD1 interaction is required to form a functional HRD1 ERAD complexREVIEWER COMMENTS

Reviewer #1 (Remarks to the Author):

This is an interesting study on the function of SEL1L-HRD1 interaction in ERAD and the impairment in mice carrying a variant SEL1LS658P . It includes extensive well devised experimental data. The following concerns should be addressed:

There are no page or line numbers, which would facilitate the review. I will copy fragments of the article when necessary:

1) "These data suggest that SEL1L S658P causes a physical collision between SEL1L F668 and HRD1 Y30, thereby attenuating SEL1L-HRD1 interaction." A major concern is that to have a definitive proof of this, mutations in HRD1 Y30 should be made and tested. Alternatively, the conclusions should be moderated (here and in the Abstract and Discussion), speculating that the results are consistent with this.

2) Abstract "definitive evidence for the importance of SEL1L in HRD1 ERAD is lacking. " This statement is exaggerated as there are many previous studies on this topic.

3) Abstract "...causes HRD1 dysfunction by generating electrostatic repulsion between SEL1L F668 and HRD1 Y30 residues. " This is not definitely proven, as I comment above, it should be moderated.

4) Abstract "...but the E2 UBE2J1 and retrotranslocon DERLIN, to HRD1." The results show Derlin2, which in any case has not been formally shown to be a part of a retrotranslocon, so perhaps "putative retrotranslocon component". This correction should also be introduced in other parts of the text when the protein is mentioned.

5) In the second paragraph of the Introduction, several additional important ERAD components should be mentioned such as p97 (VCP), Herp (HERPUD1), etc.

6) "These data show that SEL1LS658P KI mice exhibit growth retardation, and signs of early onset non-progressive ataxia, establishing the disease-causality and pathogenicity of this allele. " It should be noted that the effects are mild.

7) "Hence, these data show that KI mice exhibit microcephaly, with no massive cell death or inflammation in the brain." It should be noted that the effect is mild.

8) "In line with our previous study that UPR sensor IRE1 α is an ERAD substrate 23, its protein level was increased by ~4 folds; however, neither IRE1 α phosphorylation nor splicing of its downstream effector Xbp1 mRNA was elevated..." There should be a comment on this in the results or discussion. How can it be explained that there is no increased XBP1 splicing with such an increase in IRE1?

9) Fig. 3b, why is total PERK much increased? Comment on this.

10) "...which was attenuated upon the over-expression of SEL1L WT (lane 3-5 vs. 2), but not SEL1L S658P (lane 6-8 vs. 3-5, Fig. 4a and quantitated in Fig. 4b)." There is still some effect, perhaps "but much less SEL1L S658P".

11) Section "Sequence and structural analyses of SEL1LS658P variant" analyzes the predictions on S658P proximity to HRD1 loops but does not comment on the proximity to OS-9. In light of this the results that show no effect of the mutation on SEL1L interaction with OS-9 might be unexpected, so the proximity to OS-9 should be mentioned in this section and the lack of interference commented in the Discussion.

12) Fig. 7d: It seems that the names Group I and Group II were swapped here.

13) "...the upregulation of ER chaperones to increase folding efficiency, enhanced aggregation and sequestration of misfolded proteins to hence attenuate proteotoxicity and the activation of ER-phagy to clear misfolded protein aggregates in the ER 62,63." It could be added here that perhaps in vivo there is compensation by ERAD involving alternative E3 ligases.

14) The English is good in general, but I spotted several typos, such as "signification subfraction", "Structral predication", "whther the resuced interacion". The text should be carefully edited.

Reviewer #3 (Remarks to the Author):

In this manuscript, the authors explore the consequences of a disease-associated mutation in SEL1L, recapitulating its effects in vivo to analyse phenotypic effects and subsequently delve into the molecular

mechanics of the observed effects in cellular models. They highlight intriguing features of the transgenic animals caused by the mutation and link them to the defect in ERAD, as demonstrated in their cellular model. The authors underscore the potential selective vulnerability of Purkinje cells to ERAD disruption and the developmental nature of the relevant pathology. The adept use of predictive structure analysis via AlphaFold has enabled the authors to deepen their understanding of the consequences of SEL1L mutation, which they further validated through co-IP interaction analysis upon SEL1L mutagenesis. The manuscript also characterizes the relevant ER stress status associated with SEL1L defect. Integrating their results, the authors suggest a potential selective vulnerability of Purkinje cells to ERAD perturbation, hypothesising the existence of a specific ERAD substrate that exposes these cells to SEL1L malfunction.

Overall, the conclusions appear to be drawn from comprehensive, high-quality experimental data and represent a significant advancement in understanding the protein's role in ERAD and the implications of its malfunction for brain health. This information will be valuable for the field and broadly as it highlights and mechanistically describe the connection between neuronal health and ERAD inviting further elucidation of this process.

A few points the authors should consider:

- In Fig. 4, the effect of the SEL1L mutation on the degradation rate of different substrates in the cycloheximide chase appears to be particularly prominent for CD147 as an ERAD substrate. The differences for various substrates should be contextualised with their overall half-lives. Given this substrate's sensitivity to SEL1L activity, experiments showing the compensatory effect of the SEL1L S658P/F668Y variant should be presented. This would substantiate the conclusions drawn from the observed partial reversal of HMW accumulation, which isn't a direct ERAD-reporting observation.
- Supported by the structural analysis, in Fig. 6, the authors investigate how SEL1LS658P attenuates the SEL1L-HRD1 interaction and attribute this to the interface between F668 (SEL1L) and Y30 (HRD1). While the results back their hypothesis, additional evidence would solidify this conclusion. Demonstrating that mutating HRD1, targeting Y30, produces a similar effect would be pivotal. As the co-IP method doesn't conclusively indicate the directness of interactions, these further evidences would be needed for establishing the effect on SEL1-HRD1 interaction. Affinity measurements between the variants of the two proteins would also provide a more definitive support for this core finding.
- In Extended Data Fig. 6c, the authors suggest that the asparagine mutant (F668N) doesn't disrupt the interaction between SEL1L and HRD1. Given that asparagine isn't an aromatic residue, the authors should discuss these results in the context of their aromatic-aromatic interaction theory.
- The authors should consider elaborating on the rationale behind defining the phenotype as ataxia.

Minor Corrections:

- Page 7: Change "ration" to "ratio?"
- Page 9: Correct "whether" and "rescued."
- Should the F668 mutant that counteracts the perturbation by the S658P mutation be "F668Y" instead of "F668W?"
- In Fig. 7d, should the group labels be switched? Should the purple be Group I, and the blue be Group II?
- In Fig. 7c, the heatmap for DERL2 in the SEL1L IP panel seems inconsistent with other data. The authors should address the potential reasons for DERL2's absence in mass spectrometry.

Reviewer #4 (Remarks to the Author):

I understand the authors' efforts in trying to show the physiological importance of interaction between Sel1L and HRD1 by using the pathological mutation SEL1L(S658P) identified in dog suffering cerebellar ataxia. They generated SEL1L(S658P) knock-in (KI) mice and analyzed the cerebellar ataxia phenotype of homozygous KI mice in detail. The importance of the SEL1L(S658P) mutation was evident from genetic analysis in dog lines, the data obtained from the SEL1L(S658P) knock-in mice is only confirmatory. On the other hand, they showed that SEL1L(S658P) mutation reduced the interaction with HRD1 in HEK293 cells and hypothesized that the attenuation of SEL1L-HRD1 interaction causes HRD1 dysfunction by creating a repulsion between SEL1L(F668) and HRD1(Y30) residues in silico. However, there is no physiological relevance between the data obtained from mice experiments and those from HEK293 cells or in silico. Their claim that the SEL1L(S658P) mutation causes only marginal ER stress and does not activate the caspase pathway. However, it is not clear whether ERAD is inhibited or HRD1 function is impaired in the cerebellum of the SEL1L(S658P) KI mouse. The two topics of the manuscript (the interaction of Sel1L with HRD1 in HEK293 cells and the pathological mechanisms of the SEL1L(S658P) KI mouse phenotypes) are not well integrated, and the manuscript would be better presented as two separate papers.

Other points

It is unclear why a half of SEL1L(S658P) homozygous KI mice are lethal.

In the beam walking test, quantitative data on the number of times foot slipped is needed.

P6. Not shown data should be shown.

Since CHOP is considerably elevated in the cerebellum of Finnish Hounds lines (ref 47), this should be carefully examined in KI mice as well.

They claimed that neurons including Purkinje cells adapt to the expression of SEL1L(S658P) without eliciting an overt ER stress or cell death (P7). However, because the cerebellum contains many Bergmann glia and other cells, more detailed studies must be carefully conducted to confirm Purkinje cell-specific changes.

We thank all three reviewers for their insightful and constructive comments. We now have carefully addressed all the comments from the reviewers, which have been instrumental to further improve our manuscript.

Reviewer #1 (Remarks to the Author):

This is an interesting study on the function of SEL1L-HRD1 interaction in ERAD and the impairment in mice carrying a variant SEL1LS658P. It includes extensive well devised experimental data. The following concerns should be addressed:

There are no page or line numbers, which would facilitate the review. I will copy fragments of the article when necessary:

We have now added the page and line numbers in the manuscript.

1) "These data suggest that SEL1L S658P causes a physical collision between SEL1L F668 and HRD1 Y30, thereby attenuating SEL1L-HRD1 interaction." A major concern is that to have a definitive proof of this, mutations in HRD1 Y30 should be made and tested. Alternatively, the conclusions should be moderated (here and in the Abstract and Discussion), speculating that the results are consistent with this.

We thank the reviewer for this great comment. As suggested, we have now generated HRD1 Y30A, Y30D, Y30K and Y30F mutants and assessed their impact on HRD1 interaction with SEL1L and other ERAD components. Our IP data showed that HRD1-Y30 mutated to A, D or K significantly disrupted the SEL1L-HRD1 interaction by approximately 80-90%, while Y30F resulted in a 30% reduction (**Response Figure 1**). The disruption of SEL1L-HRD1 interaction abolished the interaction of HRD1 with the ER lectin OS9, while having no impact on the interaction with FAM8A1 (**Response Figure 1**). This data suggests that the HRD1 Y30 is critical determinant for the SEL1L-HRD1 interaction and complex formation. In addition, we have tone down the conclusion throughout the text. This data is now shown in **Extended Data Fig. 7c** in the revised manuscript.

Response Figure 1. HRD1 Y30 is critical for the SEL1L-HRD1 interaction. Immunoprecipitation of FLAG-agarose in *HRD1*^{-/-} HEK293T cells transfected with indicated HRD1-FLAG constructs to exam the interaction with SEL1L, OS9 and FAM8A1, with quantitation shown below the gel as means from two independent repeats.

Result: "We first examine whether SEL1L F668 or HRD1 Y30 is critical for the SEL1L-HRD1 interaction. Indeed, HRD1 Y30 mutated to Ala (A), Asp (D) or (K) significantly disrupted the SEL1L-HRD1 interaction by 80-90%, and 30% when mutated to Phe (F). The disruption of SEL1L-HRD1 interaction abolished the interaction of HRD1 with OS9, while having no impact on the interaction with FAM8A1."

Result: "These data suggested that SEL1L S658P may cause a physical collision between SEL1L and HRD1 via SEL1L F668 and HRD1 Y30 residues, while having no effect on SEL1L-OS9/ERLEC1 interaction. However, the definitive support for this model will require the affinity measurements between the two proteins."

2) Abstract "definitive evidence for the importance of SEL1L in HRD1 ERAD is lacking. "This statement is exaggerated as there are many previous studies on this topic.

We thank the reviewer for this great comment. We now have changed this statement in the Abstract.

Abstract: "in vivo evidence for the importance of SEL1L in the ERAD complex formation and its (patho-)physiological relevance in mammals remains limited."

3) Abstract "...causes HRD1 dysfunction by generating electrostatic repulsion between SEL1L F668 and HRD1 Y30 residues. "This is not definitely proven, as I comment above, it should be moderated.

See response to Point 1 and **Response Figure 1**.

4) Abstract "...but the E2 UBE2J1 and retrotranslocon DERLIN, to HRD1." The results show Derlin2, which in any case has not been formally shown to be a part of a retrotranslocon, so perhaps "putative retrotranslocon component". This correction should also be introduced in other parts of the text when the protein is mentioned.

We thank the reviewer for this great comment. We agree with that DERLIN2, as well as other DERLIN proteins in mammals, has not been formally proved as a retrotranslocon. We now have updated the text throughout in the revised manuscript as suggested.

5) In the second paragraph of the Introduction, several additional important ERAD components should be mentioned such as p97 (VCP), Herp (HERPUD1), etc.

We thank the reviewer for this great comment. We have now included additional factors such as p97(VCP) and Herp (HERPUD1) in the Introduction.

Introduction: "...subsequently excised from the ER membrane by the cytosolic AAA-ATPase Cdc48/VCP for proteasomal degradation. Human HERP (Usa1p in yeast) promotes HRD1 oligomerization as well as the formation of ERAD complex in vitro, which is required for substrate retrotranslocation."

6) "These data show that SEL1LS658P KI mice exhibit growth retardation, and signs of early onset non-progressive ataxia, establishing the disease-causality and pathogenicity of this allele. "It should be noted that the effects are mild.

We thank the reviewer for this comment. We have now revised it in the text and pasted below.

Result: "These data show that SEL1L^{S658P} KI mice exhibit mild growth retardation, and signs of early onset non-progressive mild ataxia, establishing the disease-causality and pathogenicity of this allele."

7) "Hence, these data show that KI mice exhibit microcephaly, with no massive cell death or inflammation in the brain." It should be noted that the effect is mild.

We thank the reviewer for this comment. We have now revised it on Page 6 and pasted below.

Result: "Hence, these data show that KI mice exhibit mild microcephaly, with no massive cell death or inflammation in the brain."

8) "In line with our previous study that UPR sensor IRE1 α is an ERAD substrate 23, its protein level was increased by ~4 folds; however, neither IRE1 α phosphorylation nor splicing of its downstream effector Xbp1 mRNA was elevated..." There should be a comment on this in the results or discussion. How can it be explained that there is no increased XBP1 splicing with such an increase in IRE1?

We thank the reviewer for this great comment and apologize for the confusion. The IRE1 α protein levels in the KI cerebellum were increased by ~4 folds (**Response Figure 2a**), in line with the notion that UPR sensor IRE1 α is an HRD1-SEL1L ERAD substrate and ERAD deficiency causes IRE1 α protein stabilization and accumulation¹. However, IRE1 α phosphorylation was not elevated as shown in **Response Figure 2a** and consistently, its downstream effector Xbp1 mRNA splicing was not significantly increased in the KI cerebellum (**Response Figure 2b**). We now have added a positive control (Liver+TM) for Xbp1 mRNA splicing as shown

in **Response Figure 2b**. This data suggested that the SEL1L^{S658P} KI cerebellum does not instigate an overt IRE1 α activation. **Response Figure 2** is now shown in **Fig. 4a** and **4d**. We have now commented it in the Discussion and pasted below. The most likely explanation for no increased Xbp1 splicing or UPR activation is cellular adaptation. We propose that there is an activation of adaptive mechanisms in response to the expression ERAD variants, such as the upregulation of ER chaperones GRP94, PDI and BiP to increase folding efficiency (**Response Figure 2c**) and/or ER-phagy to degrade misfolded protein aggregates in the absence of SEL1L-HRD1 ERAD as we recently shown in ERAD KO cells/tissues^{2,3}.

Response Figure 2. Lack of an overt UPR in the cerebellum of SEL1L^{S658P} mice. (a-b) Western blot analysis of IRE1 α phosphorylation (a) and RT-PCR of Xbp1 splicing (b) in the cerebellum of 5-week-old mice. 0/p, non-/phosphorylation; u/s, un-/spliced Xbp1. Quantitation of IRE1 α and the ratio of spliced to total Xbp1 shown below the gel as mean (n = 4-5 mice per group). TM, tunicamycin (ER stress inducer). **(c)** Western blot analysis of ER chaperones GRP94, BiP and PDI in the

cerebellum of 5-week-old mice with quantitation shown on the right (n = 4-8 mice per group). Values, mean ± SEM. *p<0.05, **p<0.01 and ***p<0.001 by two-tailed Student's *t*-test (c).

Discussion: “The lack of an overt UPR in the KI mice is likely due to various adaptive mechanisms in response to a hypomorphic variant, including, but not limited to, the upregulation of ER chaperones to increase folding efficiency, enhanced aggregation and sequestration of misfolded proteins to hence attenuate proteotoxicity, the activation of ER-phagy to clear protein aggregates, and/or ERAD involving alternative E3 ligase to clear misfolded proteins in the ER.”

9) Fig. 3b, why is total PERK much increased? Comment on this.

We thank the reviewer for this great comment. The reason underlying the increased PERK in the KI cerebellum is currently unclear, although it is likely caused by the elevated *PERK* gene transcription.

10) “...which was attenuated upon the over-expression of SEL1L WT (lane 3-5 vs. 2), but not SEL1L S658P (lane 6-8 vs. 3-5, Fig. 4a and quantitated in Fig. 4b).” There is still some effect, perhaps “but much less SEL1L S658P”.

We thank the reviewer for this great comment and have revised the text as suggested.

Results: “In line with these findings, turnover of a known ERAD substrate, a disease mutant of pro-arginine vasopressin (proAVP) at residue 57 (Gly-to-Ser, Gly57Ser), was attenuated in SEL1L^{S658P} transfected SEL1L^{-/-} HEK293T cells, leading to its accumulation and the formation of HMW aggregates (lane 6-8 vs. 3-5, Fig. 5h and quantitated in Fig. 5i).”

11) Section “Sequence and structural analyses of SEL1LS658P variant” analyzes the predictions on S658P proximity to HRD1 loops but does not comment on the proximity to OS-9. In light of this the results that show no effect of the mutation on SEL1L interaction with OS-9 might be unexpected, so the proximity to OS-9 should be mentioned in this section and the lack of interference commented in the Discussion.

We thank the reviewer for this great comment. We have now added a side view of the SEL1L-OS9 interface around the SEL1L 658 position. Indeed, as suggested by the reviewer, SEL1L E659 and R655 may interact with OS9 (**Response Figure 3c**). However, further immunoprecipitation and biochemical data showed that SEL1L S658P mutation does not affect its interaction with OS9 and ERLEC1 (Fig. 6e) for the unknown reason.

We have made changes accordingly in the revised manuscript to reflect these findings. **Response Figure 3** is now shown in **Fig. 6c-d** and **Extended Data Fig. 6c** in the revised manuscript. We also added a discussion on this point in the Results.

Response Figure 3. Structural prediction of human SEL1L/OS9/HRD1/DERLIN ERAD complex. (a) Structural prediction of human SEL1L/OS9/HRD1/DERLIN ERAD complex using AlphaFold2 with SEL1L S658 shown in pink. The arrow indicates the amphipathic helix which interacts with HRD1. **(b-c)** Side views of a space-filling model of the HRD1-SEL1L interface containing TM1-2 of HRD1 and the amphipathic helix of SEL1L **(b)**, and the SEL1L-OS9 interface around the SEL1L S658 position, with the dotted lines indicating SEL1L-OS9 interaction residues **(c)**.

Results: “On the other hand, SEL1L S658 is in proximity to residues E659 and R655 of SEL1L, which may be involved in the interaction with OS9.”

Result: “These data suggested that SEL1L S658P may cause a physical collision between SEL1L and HRD1 via SEL1L F668 and HRD1 Y30 residues, while having no effect on SEL1L-OS9/ERLEC1 interaction. However, the definitive support for this model will require the affinity measurements between the two proteins.”

12) Fig. 7d: It seems that the names Group I and Group II were swapped here.

We thank the reviewer for this comment and have now corrected Group I and II in the Figure.

13) "...the upregulation of ER chaperones to increase folding efficiency, enhanced aggregation and sequestration of misfolded proteins to hence attenuate proteotoxicity and the activation of ER-phagy to clear misfolded protein aggregates in the ER 62,63." It could be added here that perhaps in vivo there is compensation by ERAD involving alternative E3 ligases.

We thank the reviewer for this great comment and have included the Discussion on the alternative E3 ligases and pasted blow.

Discussion: *"The lack of an overt UPR in the KI mice is likely due to various adaptive mechanisms in response to a hypomorphic variant, including, but not limited to, the upregulation of ER chaperones to increase folding efficiency, enhanced aggregation and sequestration of misfolded proteins to hence attenuate proteotoxicity, the activation of ER-phagy to clear protein aggregates, and/or ERAD involving alternative E3 ligase to clear misfolded proteins in the ER."*

14) The English is good in general, but I spotted several typos, such as "signification subfraction ", "Structral predication ", "whthter the resuced interacion". The text should be carefully edited.

We thank the reviewer for pointing out those typos. We have now carefully edited the text and corrected typos.

Reviewer #3 (Remarks to the Author):

In this manuscript, the authors explore the consequences of a disease-associated mutation in SEL1L, recapitulating its effects in vivo to analyse phenotypic effects and subsequently delve into the molecular mechanics of the observed effects in cellular models. They highlight intriguing features of the transgenic animals caused by the mutation and link them to the defect in ERAD, as demonstrated in their cellular model. The authors underscore the potential selective vulnerability of Purkinje cells to ERAD disruption and the developmental nature of the relevant pathology. The adept use of predictive structure analysis via AlphaFold has enabled the authors to deepen their understanding of the consequences of SEL1L mutation, which they further validated through co-IP interaction analysis upon SEL1 mutagenesis. The manuscript also characterizes the relevant ER stress status associated with SEL1L defect. Integrating their results, the authors suggest a potential selective vulnerability of Purkinje cells to ERAD perturbation, hypothesising the existence of a specific ERAD substrate that exposes these cells to SEL1 malfunction. Overall, the conclusions appear to be drawn from comprehensive, high-quality experimental data and represent a significant advancement in understanding the protein's role in ERAD and the implications of its malfunction for brain health. This information will be valuable for the field and broadly as it highlights and mechanistically describe the connection between neuronal health and ERAD inviting further elucidation of this process.

We thank this reviewer for the constructive comments.

A few points the authors should consider:

- In Fig. 4, the effect of the SEL1L mutation on the degradation rate of different substrates in the cycloheximide chase appears to be particularly prominent for CD147 as an ERAD substrate. The differences for various substrates should be contextualised with their overall half-lives. Given this substrate's sensitivity to SEL1L activity, experiments showing the compensatory effect of the SEL1L S658P/F668Y variant should be presented. This would substantiate the conclusions drawn from the observed partial reversal of HMW accumulation, which isn't a direct ERAD-reporting observation.

We thank the reviewer for this great comment. Our data showed that SEL1L S658P/F668Y can partially rescue the defects of ERAD dysfunction towards a model substrate when overexpressed (**Response Figure 4a**). However, our effort to generate SEL1L S658P/F668Y knock-in HEK293T cells failed as the cells exhibited either resistance to the Cas9 electroporation or stopped growing following electroporation. In SEL1L KO HEK293T cells with overexpressed SEL1L (double) mutants, we noted that the overexpressed SEL1L, both WT and mutants, were unable to rescue ERAD dysfunction as measured by the levels of endogenous substrates in *SEL1L^{-/-}* cells *in vitro* (**Response Figure 4b**). This was also true when cells transfected with non-tagged SEL1L WT or mutants (**Response Figure 4c**). While we are currently exploring alternative approaches to address the issue raised by the reviewer, we have toned down the conclusion for this part in the revised manuscript (see the revised text below), due to the lack of data for the S658P/F668Y variant.

Results: “These data suggested that SEL1L S658P may cause a physical collision between SEL1L and HRD1 via SEL1L F668 and HRD1 Y30 residues, while having no effect on SEL1L-OS9/ERLEC1 interaction. However, the definitive support for this model will require the affinity measurements between the two proteins.”

Response Figure 4. Overexpressed SEL1L, either WT or mutant, is non-functional towards endogenous ERAD substrates in SEL1L^{-/-} HEK293T cells. (a) Reducing and non-reducing SDS-PAGE and Western blot analyses of proAVP-G57S high molecular-weight (HMW) aggregates in WT or SEL1L^{-/-} HEK293T cells transfected with indicated SEL1L-FLAG constructs (n = 4 independent samples for each genotype). **(b-c)** Western blot analysis of known endogenous substrates (IRE1α, OS9 and CD147) in WT and/or SEL1L^{-/-} HEK293T cells transfected with indicated SEL1L-FLAG at different doses of SEL1L constructs (b) or

different SEL1L constructs (c) (two independent repeats).

• Supported by the structural analysis, in Fig. 6, the authors investigate how SEL1LS658P attenuates the SEL1L-HRD1 interaction and attribute this to the interface between F668 (SEL1L) and Y30 (HRD1). While the results back their hypothesis, additional evidence would solidify this conclusion. Demonstrating that mutating HRD1, targeting Y30, produces a similar effect would be pivotal. As the co-IP method doesn't conclusively indicate the directness of interactions, these further evidences would be needed for establishing the effect on SEL1-HRD1 interaction. Affinity measurements between the variants of the two proteins would also provide a more definitive support for this core finding.

We thank the reviewer for this great comment. As suggested, we have now generated HRD1 Y30A, Y30D, Y30K and Y30F mutants and assessed their impact on HRD1 interaction with SEL1L and other ERAD components. Our IP data showed that HRD1-Y30 mutated to A, D or K significantly disrupted the SEL1L-HRD1 interaction by approximately 80-90%, while Y30F retained ~ 70% interaction (**Response Figure 5**). The disruption of SEL1L-HRD1 interaction also abolished the interaction of HRD1 with the ER lectin OS9, while having no impact on the interaction with FAM8A1 (**Response Figure 5**). This data suggests that the HRD1 Y30 is critical determinant for the SEL1L-HRD1 interaction. This data is now shown in **Extended Data Fig. 7c** in the revised manuscript. We were not able to perform affinity measurements due to the technical challenges, while fragments of SEL1L and HRD1 are known to directly interact *in vitro*⁴. Hence, we now have toned down our conclusions on the direct interaction.

Result: “We first examine whether SEL1L F668 or HRD1 Y30 is critical for the SEL1L-HRD1 interaction. Indeed, HRD1 Y30 mutated to Ala (A), Asp (D) or (K) significantly disrupted the SEL1L-HRD1 interaction by 80-90%, and 30% when mutated to Phe (F). The disruption of SEL1L-HRD1 interaction abolished the interaction of HRD1 with OS9, while having no impact on the interaction with FAM8A1.”

Result: “These data suggested that SEL1L S658P may cause a physical collision between SEL1L and HRD1 via SEL1L F668 and HRD1 Y30 residues, while having no effect on SEL1L-OS9/ERLEC1 interaction. However, the definitive support for this model will require the affinity measurements between the two proteins.”

Response Figure 5. HRD1 Y30 is critical for the SEL1L-HRD1 interaction. (a) Immunoprecipitation of FLAG-agarose in HRD1^{-/-} HEK293T cells transfected with indicated HRD1-FLAG constructs to examine the interaction with SEL1L, OS9 and FAM8A1, with quantitation shown below the gel as means from two independent repeats.

• In Extended Data Fig. 6c, the authors suggest that the asparagine mutant (F668N) doesn't disrupt the interaction between SEL1L and HRD1. Given that asparagine isn't an aromatic residue, the authors should discuss these results in the context of their aromatic-aromatic interaction theory.

We thank the reviewer for this great comment. We have now commented the result of F668N in the Results and pasted below.

Results: *“Moreover, F668 mutated to Asn (N), with a carboxamide side chain, had no effect on the SEL1L-HRD1 interaction.”*

• The authors should consider elaborating on the rationale behind defining the phenotype as ataxia. We thank the reviewer for this great comment. The rationale was the phenotype of the KI mice being consistent with the clinical manifestations in the Finnish Hound suffering cerebellar ataxia ⁵.

Discussion: *“Here we show that impaired SEL1L-HRD1 interaction is sufficient to drive disease pathogenesis including partial embryonic lethality, developmental delay, microcephaly and early-onset ataxia in mice as shown in the affected Finnish Hounds as previously reported ⁵.”*

Minor Corrections:

- Page 7: Change "ration" to "ratio?"
- Page 9: Correct "whether" and "rescued."
- Should the F668 mutant that counteracts the perturbation by the S658P mutation be "F668Y" instead of "F668W?"
- In Fig. 7d, should the group labels be switched? Should the purple be Group I, and the blue be Group II? We thank the reviewer for pointing out these typos and have fixed them as suggested.

• In Fig. 7c, the heatmap for DERL2 in the SEL1L IP panel seems inconsistent with other data. The authors should address the potential reasons for DERL2's absence in mass spectrometry.

We thank the reviewer for this great comment. We consistently saw the interaction when we overexpress SEL1L, but not using the endogenous SEL1L IP. We speculated that these discrepancies are likely due to the low affinity of the interaction and/or abundance of the proteins.

Reviewer #4 (Remarks to the Author):

I understand the authors' efforts in trying to show the physiological importance of interaction between Sel1L and HRD1 by using the pathological mutation SEL1L(S658P) identified in dog suffering cerebellar ataxia. They generated SEL1L(S658P) knock-in (KI) mice and analyzed the cerebellar ataxia phenotype of homozygous KI mice in detail. The importance of the SEL1L(S658P) mutation was evident from genetic analysis in dog lines, the data obtained from the SEL1L(S658P) cock-in mice is only confirmatory. On the other hand, they showed that SEL1L(S658P) mutation reduced the interaction with HRD1 in HEK293 cells and hypothesized that the attenuation of SEL1L-HRD1 interaction causes HRD1 dysfunction by creating a repulsion between SEL1L(F668) and HRD1(Y30) residues in silico. However, there is no physiological relevance between the data obtained from mice experiments and those from HEK293 cells or in silico. Their claim that the SEL1L(S658P) mutation causes only marginal ER stress and does not activate the caspase pathway. However, it is not clear whether ERAD is inhibited or HRD1 function is impaired in the cerebellum of the SEL1L(S658P) KI mouse. The two topics of the manuscript (the interaction of Sel1L with HRD1 in HEK293 cells and the pathological mechanisms of the SEL1L(S658P) KI mouse phenotypes) are not well integrated, and the manuscript would be better presented as two separate papers.

We thank the reviewer for these constructive comments. To address the question that whether ERAD is inhibited or HRD1 function is impaired in the cerebellum of SEL1L KI mice, we have performed immunofluorescence staining of SEL1L in the cerebellum and Western blot analysis of SEL1L-HRD1 ERAD (**Response Figure 6**). Confocal microscopic analysis revealed that SEL1L is highly enriched in the ER of Purkinje cells in the cerebellum (**Response Figure 6a-b**), indicating Purkinje cells may have highly active SEL1L-HRD1 ERAD. Moreover, our Western blot analysis showed that SEL1L and HRD1 protein levels are reduced by 20% and 60%, respectively, in the KI cerebellum (**Response Figure 6c**). On the other hand, IRE1 α and CD147, two known SEL1L-HRD1 ERAD substrates ^{1,6}, are increased in the KI cerebellum (**Response Figure 6c-e**). These data indicate that SEL1L^{S658P} impairs HRD1 protein stability and ERAD function in the KI cerebellum, which is further supported by our biochemical data performed in HEK293T cells

(Fig. 5). Response Figure 6 is now shown in Fig. 4a, 5a-c and Extended Data Fig 4 of the revised manuscript.

Results: “Confocal microscopic analysis revealed that SEL1L is highly enriched in the ER of Purkinje cells in the cerebellum. Western blot analysis showed that SEL1L and HRD1 protein levels were reduced by 20 and 60%, respectively, in the cerebellum of 5-week-old KI mice relative to those in WT littermates. These changes were associated with the accumulation of two known ERAD substrates IRE1α and CD147.”

Response Figure 6. SEL1L highly enrich in the ER of Purkinje cells in the cerebellum. (a-b) Representative confocal images of SEL1L (green) and KDEL (ER marker, red) in the whole brain (a) and the Purkinje cells (b) of 5-week-old mice (two independent repeats). (c-d) Western blot analysis of SEL1L, HRD1 and known ERAD substrate CD147 in the cerebellum of 5-week-old mice with quantitation shown in (d) (n = 3 mice per group). (e) Western blot analysis of known ERAD substrate IRE1α in the cerebellum of 5-week-old mice with quantitation shown below the gel as mean. Values, mean ± SEM. **p<0.01 and ****p<0.0001 by two-tailed Student’s t-test (d).

Other points

It is unclear why a half of SEL1L(S658P) homozygous KI mice are lethal.

We thank the reviewer for this great comment. In mammals, global deletion of Sel1L causes embryonic lethality⁷. The reason that a half of SEL1L KI mice is lethal remains unclear to us. One possible explanation could be the genetic background of the pups – the founders of these SEL1L S658P mice were on the B6/SJL mixed background. We now have included all these information in the Methods.

Method: “A mixture of Cas9 protein (Sigma), sgRNAs, and donor DNA was microinjected into fertilized mouse eggs on the B6/SJL background. The injected zygotes were then transferred into pseudopregnant females. The founders were then bred separately to WT C57BL6/J mice to obtain F1 heterozygous SEL1L^{S658P} KI mice. F1 heterozygous SEL1L^{S658P} mice were inter-crossed to generate homozygous SEL1L^{S658P} KI mice and its WT and heterozygous littermates.”

In the beam walking test, quantitative data on the number of times foot slipped is needed.

We thank the reviewer for this great comment and have provided the quantitation in Fig. 2g (Response Figure 7).

Response Figure 7. Quantitation of the number of slips. The quantitation of the number slips of 6- and 12-week-old mice (n = 6-9 mice per group). Values, mean ± SEM. *p<0.05 by two-tailed Student’s t-test.

P6. Not shown data should be shown.

We thank the reviewer for this great comment and have included all the data in Fig. 2 (Response Figure 8). Since we don’t have HET mice data for the balance beam test (Response Figure 8c-e), we have revised the text and pasted below.

Results: “HET mice were comparable to WT littermates in hindlimb clasping, gait pattern and rotarod tests (Fig. 2a-d and 2h-i), in line with the autosomal recessive nature of the variant.”

Response Figure 8. HET mice were comparable to WT littermates in hindlimb clasping, gait pattern and rotarod tests.

(a) Clasping score quantitation of 3-9 (n = 49, 51 and 38 for WT, HET and KI) and 40-48 (n = 17, 8 and 20 mice for WT, HET and KI) -week-old littermates of both genders. **(b)** cartoon schematic of paw prints (left, b) and quantitation of gait analysis (right, b) of 6-week-old littermates of both genders (n = 14-18 mice per group). **(c-e)** Representative pictures of slips on balance beam of 6 and 12-week-old KI mice **(c)**, with quantitation of the time of balance beam test **(d)** and the number of slips **(e)** (n = 6-9 mice per group). **(f-g)** Quantitation of the time **(f)** and rpm **(g)** of rotarod test from mice at 6 weeks of age with 3 days training (n = 7, 12 and 6 mice for WT, HET and KI). Values, mean ± SEM. n.s., not significant; *p<0.05, **p<0.01, ***p<0.001 and ****p<0.0001 by one-way ANOVA followed by Tukey's post hoc test (a, b), two-tailed Student t-test (d, e) and two-way ANOVA followed by Tukey's multiple comparisons test (f, g).

Since CHOP is considerably elevated in the cerebellum of Finnish Hounds lines (ref 47), this should be carefully examined in KI mice as well.

We thank the reviewer for this great comment. We have now added Western blot and RT-PCR analyses to examine CHOP protein and mRNA levels in the cerebellum of 5-week-old WT and KI mice. As shown in Fig. 4c of the revised manuscript (Response Figure 9), we observed about 2-fold increases in both mRNA and protein levels of CHOP in cerebellum compared to WT littermates. While Purkinje cells are known to be hypersensitive to ER homeostasis^{8,9}, future studies will be needed to delineate the pathological mechanisms underlying SEL1L KI mouse phenotype.

Results: “Expression of CHOP, a downstream mediator of UPR, was mildly elevated by 2 folds in the cerebellum of 5-week-old KI mice compared to WT littermates.”

Response Figure 9. CHOP is increased in the KI cerebellum. (a-b) RT-PCR and Western blot analysis of CHOP in the cerebellum of 5-week-old mice with quantitation shown below (n = 3-6 mice per group). Values, mean ± SEM. ***p<0.001 and ****p<0.0001 by two-tailed Student's t-test (a, b).

They claimed that neurons including Purkinje cells adapt to the expression of SEL1L(S658P) without eliciting an overt ER stress or cell death (P7). However, because the cerebellum contains many Bergmann glia and other cells, more detailed studies must be carefully conducted to confirm Purkinje cell-specific changes.

We thank the reviewer for this great comment. As our new data show that in cerebellum SEL1L protein is highly enriched in the Purkinje cells as shown in Fig. 5a and Extended Data Fig. 4 of the revised manuscript (Response Figure 6a-b), we speculate that its impact in cerebellum is mainly mediated through Purkinje cells in substrate(s)-dependent manner, as we recently reported in other tissues-specific SEL1L KO mouse models^{1,10-14}. In addition to biochemical analyses of UPR markers, our TEM data showed that there was no ER dilation in 3- and 24-week-old Purkinje cells in Fig. 4e-f (Response Figure 10a-b). Moreover, we now have performed IF analyses on the other cell types including neurons (NeuN), astrocytes (GFAP), and microglia (Ibal) in the cerebellum of 5-week-old mice (Response Figure 10c). The staining patterns appeared comparable between WT and KI mice (Response Figure 10c). This was further confirmed by our Western blot analyses showing no obvious changes in the protein levels of GFAP and Ibal in the cerebellum of KI mice (Response Figure 10d), while Calbindin, the maker of Purkinje cells, was mildly reduced by 15% (Response Figure 10e). With that said, we acknowledge that these studies did not indicate functional changes of various cerebellar cell types. Response Figure 10 is now shown in Fig. 4e-f, Extended Data Fig. 3a-b and Fig. 3i.

Response Figure 10. TEM, IF and Western blot analysis of Purkinje cell and cerebellum. (a-b) Representative TEM images of Purkinje cells (outlined by red dotted lines) of 3- (a) and 24- (b) week-old mice N, nucleus; M, mitochondria. $n = 2$ mice each genotype at each age. **(c)** Representative confocal images of NeuN (green), GFAP (red) and Ibal (purple) in the cerebellum of 5-week-old mice (two independent repeats). **(d-e)** Western blot analysis of GFAP, Ibal (d) and Calbindin (e) in the cerebellum of 5-week-old mice with quantitation shown below ($n = 3$ mice per group). Values, mean \pm SEM. n.s., not significant. *** $P < 0.001$ by two-tailed Student's t -test in (d, e).

Results: "In addition, examination of other neurons, astrocytes and microglia in the cerebellum of KI mice at 5 weeks of age using immunostaining and Western blot revealed no obvious changes in these populations compared to WT littermates."

References:

- 1 Sun, S. *et al.* IRE1alpha is an endogenous substrate of endoplasmic-reticulum-associated degradation. *Nat Cell Biol* **17**, 1546-1555, doi:10.1038/ncb3266 (2015).
- 2 Shrestha, N. *et al.* Integration of ER protein quality control mechanisms defines beta cell function and ER architecture. *J Clin Invest* **133**, doi:10.1172/JCI163584 (2023).
- 3 Wu, S. A. *et al.* The mechanisms to dispose of misfolded proteins in the endoplasmic reticulum of adipocytes. *Nat Commun* **14**, 3132, doi:10.1038/s41467-023-38690-4 (2023).
- 4 Jeong, H. *et al.* Crystal structure of SEL1L: Insight into the roles of SLR motifs in ERAD pathway. *Sci Rep* **6**, 20261, doi:10.1038/srep20261 (2016).
- 5 Kyostila, K. *et al.* A SEL1L mutation links a canine progressive early-onset cerebellar ataxia to the endoplasmic reticulum-associated protein degradation (ERAD) machinery. *PLoS Genet* **8**, e1002759, doi:10.1371/journal.pgen.1002759 (2012).
- 6 Tyler, R. E. *et al.* Unassembled CD147 is an endogenous endoplasmic reticulum-associated degradation substrate. *Mol Biol Cell* **23**, 4668-4678, doi:10.1091/mbc.E12-06-0428 (2012).

- 7 Francisco, A. B. *et al.* Deficiency of suppressor enhancer Lin12 1 like (SEL1L) in mice leads to systemic endoplasmic reticulum stress and embryonic lethality. *J Biol Chem* **285**, 13694-13703, doi:10.1074/jbc.M109.085340 (2010).
- 8 Zhao, L., Rosales, C., Seburn, K., Ron, D. & Ackerman, S. L. Alteration of the unfolded protein response modifies neurodegeneration in a mouse model of Marinesco-Sjogren syndrome. *Hum Mol Genet* **19**, 25-35, doi:10.1093/hmg/ddp464 (2010).
- 9 Yang, Y. *et al.* Disruption of Tmem30a results in cerebellar ataxia and degeneration of Purkinje cells. *Cell Death Dis* **9**, 899, doi:10.1038/s41419-018-0938-6 (2018).
- 10 Ji, Y. *et al.* SEL1L-HRD1 endoplasmic reticulum-associated degradation controls STING-mediated innate immunity by limiting the size of the activable STING pool. *Nat Cell Biol* **25**, 726-739, doi:10.1038/s41556-023-01138-4 (2023).
- 11 Yoshida, S. *et al.* Endoplasmic reticulum-associated degradation is required for nephrin maturation and kidney glomerular filtration function. *J Clin Invest* **131**, doi:10.1172/JCI143988 (2021).
- 12 Zhou, Z. *et al.* Endoplasmic reticulum-associated degradation regulates mitochondrial dynamics in brown adipocytes. *Science* **368**, 54-60, doi:10.1126/science.aay2494 (2020).
- 13 Kim, G. H. *et al.* Hypothalamic ER-associated degradation regulates POMC maturation, feeding, and age-associated obesity. *J Clin Invest* **128**, 1125-1140, doi:10.1172/JCI96420 (2018).
- 14 Shi, G. *et al.* ER-associated degradation is required for vasopressin prohormone processing and systemic water homeostasis. *J Clin Invest* **127**, 3897-3912, doi:10.1172/JCI94771 (2017).

REVIEWERS' COMMENTS

Reviewer #1 (Remarks to the Author):

My concerns have been addressed. The manuscript is significantly improved.

Reviewer #3 (Remarks to the Author):

The authors satisfactorily addressed all my comments. I recommend the manuscript's publication.

Reviewer #4 (Remarks to the Author):

The authors have thoughtfully addressed my comments by conducting experiments and discussing. I find their responses compelling, particularly in bridging the gap between in vivo and in vitro data that initially raised concerns for me. Moreover, the authors' recent papers on Sel1L mutations in human disease (JCI 2023) provide substantial support for the conclusions made in this manuscript.

However, the mechanism by which Srl1L mutation-dependent ERAD dysfunction leads to canine pathology remains elusive, especially given the absence of apparent neuronal cell death. Recognizing that this matter awaits future exploration, it would be better to discuss this point carefully, exploring possible explanations based on existing knowledge. Notably, it has been suggested that mechanisms other than neuronal cell death are involved in ER stress-related brain pathology (e.g., Sugiyama *iScience* 2023). To enhance the manuscript, a thorough discussion of the pathophysiological implications of ERAD dysfunction in brain pathology could be beneficial. Addressing this point before publication holds significant implications for researchers involved in the study of ER homeostasis and neurological diseases.